# From decision to action: Detailed modelling of frog tadpoles reveals neuronal mechanisms of decision-making and reproduces unpredictable swimming movements in response to sensory signals

**Andrea Ferrario**[1]*, **Andrey Palyanov**[2], **Stella Koutsikou**[3], **Wenchang Li**[4], **Steve Soffe**[5], **Alan Roberts**[5], **Roman Borisyuk**[1,6]

**1** College of Engineering, Mathematics and Physical Sciences, University of Exeter, Exeter, United Kingdom, **2** A.P. Ershov Institute of Informatics Systems, Siberian Branch, Russian Academy of Sciences, Novosibirsk, Russia, **3** Medway School of Pharmacy, University of Kent, Chatham Maritime, United Kingdom, **4** School of Psychology and Neuroscience, University of St. Andrews, St Andrews, United Kingdom, **5** School of Biological Sciences, University of Bristol, Bristol, United Kingdom, **6** Institute of Mathematical Problems of Biology, the Branch of Keldysh Institute of Applied Mathematics, Russian Academy of Sciences, Pushchino, Russia

* A.A.Ferrario@exeter.ac.uk

**Data Availability Statement:** The code of the CNS model is available from ModelDB repository at

## Abstract

How does the brain process sensory stimuli, and decide whether to initiate locomotor behaviour? To investigate this question we develop two whole body computer models of a tadpole. The "*Central Nervous System*" (*CNS*) model uses evidence from whole-cell recording to define 2300 neurons in 12 classes to study how sensory signals from the skin initiate and stop swimming. In response to skin stimulation, it generates realistic sensory pathway spiking and shows how hindbrain sensory memory populations on each side can compete to initiate reticulospinal neuron firing and start swimming. The 3-D "*Virtual Tadpole*" (*VT*) biomechanical model with realistic muscle innervation, body flexion, body-water interaction, and movement is then used to evaluate if motor nerve outputs from the *CNS model* can produce swimming-like movements in a volume of "water". We find that the whole tadpole *VT model* generates reliable and realistic swimming. Combining these two models opens new perspectives for experiments.

## Author summary

Animals constantly receive sensory signals, make decisions and generate behaviours. We see a red light at a pedestrian crossing, stop, and only walk across at a green light. Two systems control this behaviour: the nervous system processes sensory signals and commands the musculoskeletal system to generate motor responses. Most nervous and musculoskeletal systems are too complex to be able to understand even simple behaviours step by step. To simplify the problem, we study responses to touch in young frog tadpoles. Here,

http://modeldb.yale.edu/267146. The code for the VT model developed in Sibernetic is available at https://github.com/a-palyanov/sibernetic-vt.

**Funding:** A.F. acknowledges support from the UK Engineering and Physical Sciences Research Council (EPSRC) New Investigator Award (EP/R03124X/1). The work of A.P. was performed according to the Russian Federation Government research assignment for A.P. Ershov Institute of Informatics Systems SB RAS, project FWNU-2021-0005. S.K. acknowledges support from The Physiological Society UK Research Grant award. R. B. acknowledges support from the UK Biotechnology and Biological Sciences Research Council (BBSRC): BB/L000814/1, BB/T002352/1. W.L. acknowledges support from the UK Biotechnology and Biological Sciences Research Council (BBSRC): BB/T003146. The funders had no role in study design, data collection and analysis, decision to publish, or preparation of the manuscript.

**Competing interests:** The authors have declared that no competing interests exist.

detailed information is available on 12 types of brain and spinal cord neurons controlling swimming. To explore how these neurons work, we create two biologically realistic computational models: a *CNS model* of the nervous system with approximately 2300 neurons generates motor nerve activity and is fed to a *virtual tadpole biomechanical model* of the whole-body musculoskeletal system to produce movements. Our results suggest that we understand the essence of how simple behaviour is generated. We propose that a simple sensory memory process in the brain, which extends the brief sensory nerve activity, forms the basis for a decision process. This also generates unpredictability in behaviour.

## Introduction

Animal behaviour is based on interactions between the nervous and musculoskeletal systems, and the environment. How does an animal process sensory stimuli, decide whether and how to respond, and then initiate a coordinated locomotor response? In a wide range of animals there has been progress in understanding the networks generating rhythmic locomotion [1,2] but the circuits that initiate it, are not well understood. Locomotor behaviour like swimming requires: brainstem/hindbrain integration of sensory stimuli; sensory memory; a decision to respond; a choice of direction; and avoiding simultaneous activity of antagonists [3–6]. Although in many cases the role of neuronal populations in movement control has been characterised [7–9], a precise description of how sensory information is transformed into a decision to move is unclear[10]. In deciding whether to start to walk, fly or swim, all animals evaluate their sensory situation and initiate appropriate rhythmic activity. We are now in a unique position to show how this is done.

This paper describes computer modelling of a very simple developing vertebrate with a complete neuronal circuit from sensors through a decision mechanism to locomotion. Our models include the spinal cord to generate rhythmic activity, add the brainstem and sensory pathways upstream to make decisions and then use a biomechanical body and muscles downstream to reproduce swimming-like movements. The simplicity of the hatchling *Xenopus* tadpole allows modelling at a unique level of detail compared with existing models [11,12,13].

Two systems control swimming locomotion: the nervous system processes sensory signals, makes decisions and sends commands to the musculoskeletal system, which generates the movement in response to input. There is extensive knowledge about each system but much less is known about how these two systems interact and work together. It has been suggested that the activity of neuronal circuits can only be understood by considering the nervous system, muscular-skeletal system and the environment together [14,15]. Indeed, neuronal circuits interact with the environment, receiving and processing sensory signals, producing the output to activate muscles and generate movements, and then receiving new signals from the environment etc.

Crucial insights have been gained from "whole body" modelling of lamprey swimming by combining a neuronal network model of the animal's nervous system and a 2D biomechanical model [11,16]. The neuronal lamprey models use multi-compartment spiking neurons and can generate alternating activity patterns corresponding to forward and backward locomotion as well as steering to avoid obstacles. For biomechanical modelling, the lamprey body is represented by a chain of segments with flexible joints between them. The body reproduces two-dimensional movements and describes the interaction of the lamprey viscoelastic body with fluid using 2D Navier-Stokes hydrodynamics [17].

Compared with the tadpole, the lamprey neuronal network models are generic and do not specify the neuronal network anatomy, properties and axon projections for the different neuron classes and sensory pathways for swimming initiation and stopping. Body-liquid interactions in the lamprey mechanical models are described by 2D Navier-Stokes hydrodynamic equations and this method is adequate due to the cylindrical shape of the lamprey body. The geometry of the tadpole body is more complex and modelling tadpole's swimming behaviour requires a 3D description of the body and swimming environment. We therefore use true 3D modelling, where the body is reconstructed at high-resolution from experimental measurements and includes many details: a stiff notochord, muscles distributed along the body, specific densities of the belly and other body parts, etc. The interaction of the body with the surrounding liquid is modelled by a particle-based approach using the correct physical parameters for gravity, water density and viscosity.

Frog tadpoles, one of the most numerous vertebrate animals on earth, are simple, vulnerable animals that rely on swimming to escape from predators. The simplicity of the hatchling *Xenopus* tadpole and its nervous system presents a rare opportunity for experimental and computational studies [18,19] where detailed information is available on behaviour, sensory systems, the neurons in the central nervous system (*CNS*) [9], body components and their properties as well as body-water interactions [20].

Our aim is to use computer modelling of a simple vertebrate to answer key questions which have been raised about the control of movement: How do neurons in the brainstem respond to sensory signals to generate and direct locomotor behaviour [1]? What role does sensory memory play in the decision to move [3,10]? How is the unpredictability of movement initiation and direction produced [21,22]? What are the basic theoretical principles guiding these responses [23]? The results of our "whole" animal modelling propose answers to these questions and make predictions for experimental testing.

The tadpole *CNS model* includes two skin touch sensory pathways to initiate swimming, an inhibitory head skin pressure pathway to stop it, and hindbrain populations of sensory memory neurons (*hexN*s) which prolong signals from sensory pathways and turn on reticulospinal neurons in the swimming Central Pattern Generator (CPG). We propose that hexNs play a critical role in the process of swimming initiation by providing a basic sensory memory of brief skin "touch" stimuli [9,24]. One component of the *CNS model* is a neuronal mechanism of decision making based on an accumulation of competing excitations to threshold [25–28] on each side of the body. In particular, our simulations explain experimental evidence [9] for long and variable delays between stimulation and the start of swimming as well as the side of the first body flexion. Introducing unpredictability in the side of first flexion may reduce predation because the direction of tadpole movement becomes unpredictable for a hunting predator [29].

A critical step in our modelling is to evaluate whether the motor output from the *CNS model* can produce swimming movements by building a novel 3D biomechanical computer *Virtual Tadpole* (*VT*) model of the tadpole body. This model is based on previous modelling of the undulating movements of the nematode *Caenorhabditis elegans* (*C. elegans*) [30]. The *VT model* reconstructs detailed physical and anatomical measurements of the shapes and mass distribution of organs like the notochord, muscles and belly in real tadpoles. We place the reconstructed virtual body in "water" and feed patterns of motoneuron spiking from the *CNS model* to the muscle segments of the *VT model* to drive "virtual" swimming movements. Supplementary videos demonstrate that *VT model* simulations produce realistic swimming behaviour.

The combination of *CNS* and *VT models* opens a new perspective for experimental studies. Being realistic and detailed, the *CNS model* can be fitted to match experimental recordings

from immobilised tadpoles. Motone uron spiking patterns generated by the *CNS model* can activate muscle segments of the *VT model* to test what movements are generated in water. This novel approach leads to new theoretical insights for experimental testing.

## Introductory background on tadpole swimming and its neuronal control

When a hatchling *Xenopus* tadpole is touched, it flexes unpredictably to the left or right and then swims away even if the mid- and fore-brains are disconnected (Fig 1a, 1b and 1d). Swimming stops when the head bumps into solid support and attaches using mucus from a cement gland [31]. Tadpoles swim belly down because dense ventral belly cells act as ballast [32]. Segmented muscles lie on each side of the spinal cord and a notochord acts as a longitudinal skeletal rod.

Detailed information on the neurons initiating and controlling swimming comes from paired whole-cell recordings in immobilised tadpoles with dye injections to reveal neuron anatomy [18] (overview Fig 1c–1f). Neurons are mainly arranged in longitudinal rows, have short local dendrites and longitudinal axons projecting on the same or the opposite side. Motor nerve recordings reveal swimming activity to 1ms electrical stimuli which excite touch sensory neurons in the head (*tSt*) or trunk (*RB*) skin. Sensory pathway neurons fire single spikes and transmit brief excitation to the hindbrain on both sides of the body [33]. A slow wave of excitation builds up in electrically coupled hindbrain reticulospinal descending *dIN*s until firing threshold is reached on one side of the body. The *dIN* population on that side is recruited to fire, excites motoneurons (*mn*s) and swimming starts [24,34,35]. **Note**: The population of descending interneurons (*dIN*) is distributed along the body from the hindbrain to tail. We use the abbreviation *hdIN*s to indicate the population of *dIN* located in the hindbrain.

The *hdIN*s neurons make the decision to swim and drive the other swimming CPG neurons including motoneurons (Fig 1c, 1d and 1f). They are probable homologues of brainstem neurons in zebrafish larvae [6]. We have proposed a preliminary hypothesis that: (1) the wave of excitation in hindbrain *dIN*s comes from populations of hindbrain sensory memory, extension neurons (*hexN*s) on each side of the body. The *hexN*s which are excited following skin stimulation [36] (level 4 in Fig 1c); (2) The *hexN*s then fire for about 1s as a result of proposed mutual re-excitation [24]. Variation in the firing of the *hexN*s on each side of the body leads to unpredictability in the side where the *dIN* firing threshold will be reached and the first flexion of swimming will occur. Sustained swimming then depends on self-re-excitation within the *dIN* populations on each side of the body and on rebound firing following mid-cycle inhibition from the reciprocal inhibitory *cIN*s on the opposite side [19].

As swimming continues, the cycle period lengthens and stopping can occur spontaneously as a result of the extracellular accumulation of adenosine [37] or when pressure on the head skin excites trigeminal sensory neurons (*tSp*s) [31]. These excite mid-hindbrain GABAergic inhibitory reticulospinal neurons (*MHR*s) which project caudally on both sides of the body to inhibit CPG neurons and stop swimming (Fig 1c and 1e).

## Results

### How *CNS model* generates swimming behaviour

The new *CNS model* is a network of 2,308 spiking, Hodgkin-Huxley type neurons of 12 different types (Fig 1c–1f). This model can generate a complete sequence of swimming behaviour initiated and stopped by skin sensory inputs (Fig 2) which is very similar to real tadpole behaviour in response to natural stimuli. It produces motoneuron activity patterns for swimming initiated by "touch" to the head or trunk skin which evokes brief firing in the sensory pathways (Fig 2b and 2d). Swimming can be terminated by head skin "pressure" when sensory and

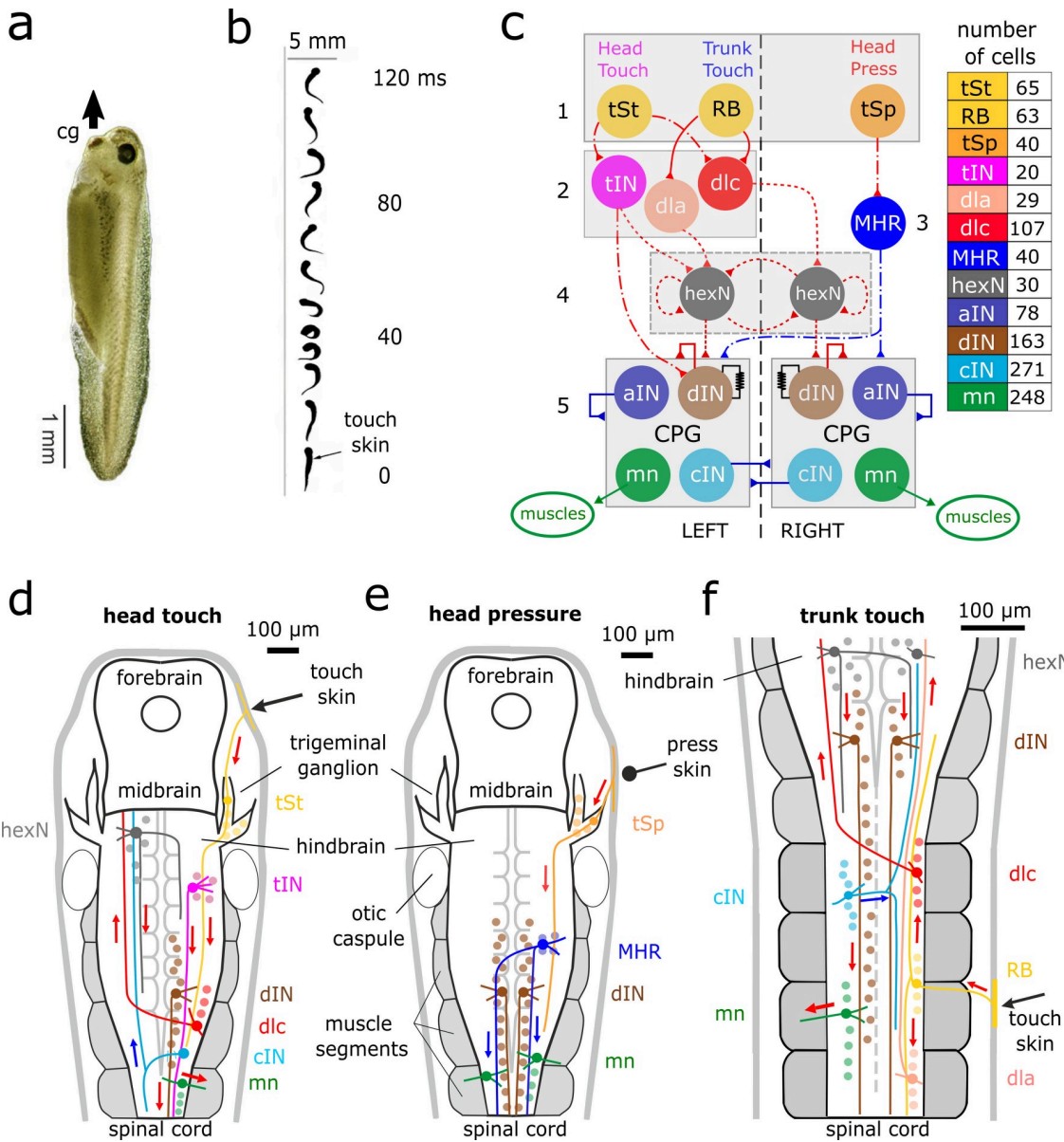

**Fig 1. Tadpole, swimming and neuronal pathways. a**. Tadpole hanging from mucus (arrow) secreted by cement gland (cg). **b**. If touched (arrow), it flexes to one side and then swims off. **c**. Functional diagram of *CNS model* network with 12 neuron types forming 5 layers: 1) skin sensory; 2) sensory projection; 3) inhibitory reticulospinal; 4) sensory memory; 5) central pattern generator. The list shows colour coding and population of each neuron type on one side. The total number of neurons in *CNS model* is 2,308 (half on each side) including 788 sensory pathway and 1,520 CPG neurons. Connections shown by solid lines have been established by "developmental" modelling based on single cell recording and dye injections [19]; dash-dotted and dotted lines indicate connections prescribed by the "probabilistic" model (dashed line means that connections probabilities are based on some limited experimental evidence). Red are excitatory and blue are inhibitory connections; resistor sign shows electrical connections. Abbreviations not defined in text: Interneurons: *aIN* = ascending; *cIN* = commissural, *tIN* = trigeminal; *dla* = dorsolateral ascending; *dlc* = dorsolateral commissural; *mn* = motoneuron; *dIN* = descending interneurons. **d-f**. Diagrams of the brain and spinal cord illustrating the neuronal pathways from head (**d**) or trunk touch (**f**) to start and from head pressure (**e**) to stop swimming. Only selected neurons are shown to make the signal propagation pathways clear: coloured arrows show activity propagation pathways (red = excitation; blue = inhibition). The hindbrain extends from 0–500 microns and the spinal cord from 500–3500 microns.

inhibitory reticulospinal neurons (*MHR*s) fire repetitively (Fig 2c). As swimming continues, cycle period lengthens due to depression in *dIN* NMDA component of recurrent connections (see also section "OPERATION OF CNS MODEL: SWIMMING INITIATION" below). In

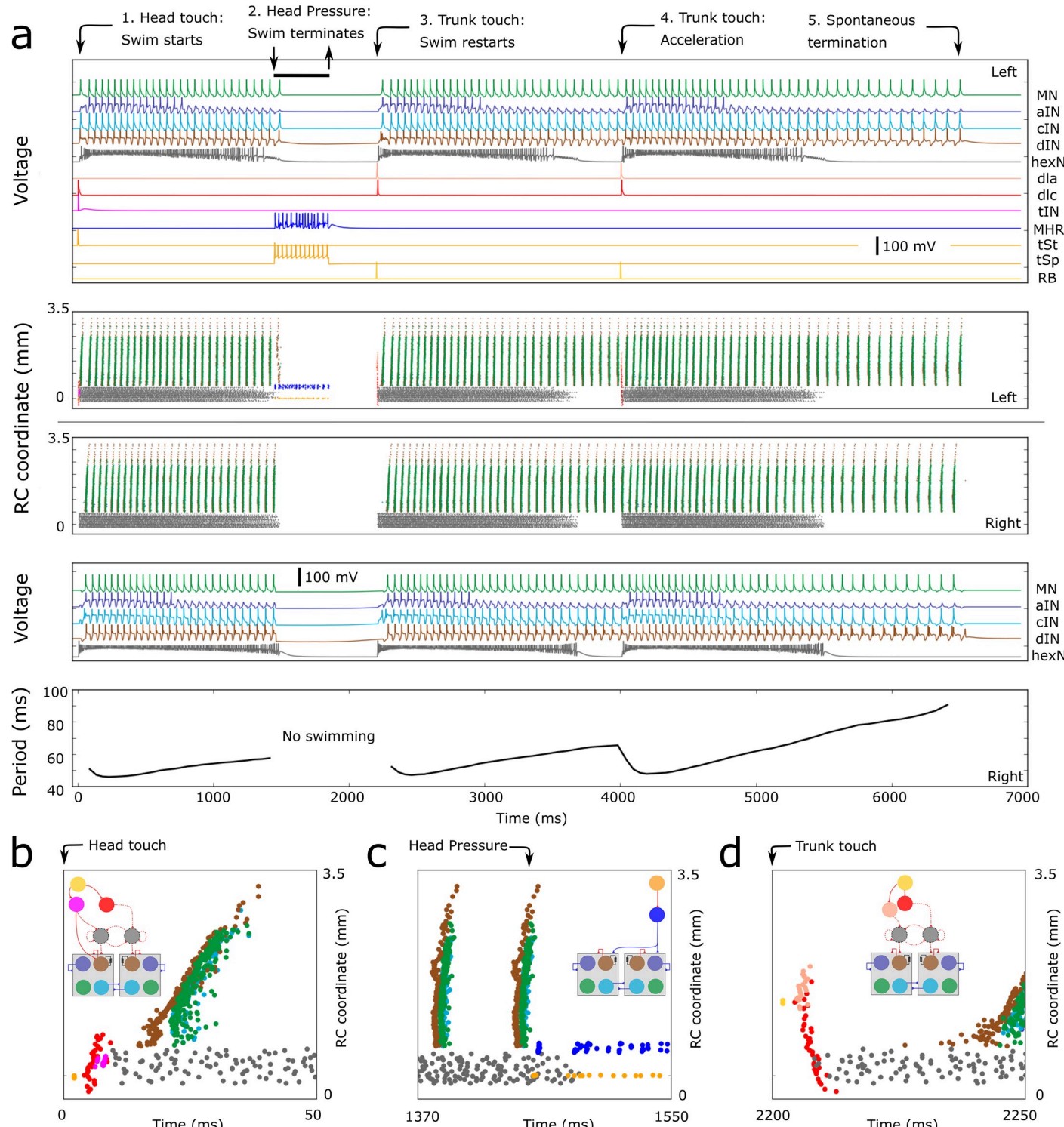

**Fig 2. Overview of *CNS model* responses to a sequence of sensory signals. a**. The "Voltage" panels show membrane potentials of selected active neurons on the left and right sides respectively. The two central panels show spike times (coloured dots, zoomed views in **b-d**) of all active neurons, the vertical coordinate is the rostro-caudal (RC) position of the spiking neuron. All current pulse sensory stimuli are applied to the left side of the body as indicated by black arrows at the top. 1) Head skin touch initiates swimming from rest (stimulus of 0.3 nA for 5ms excites 13 *tSt*s at time zero); 2) Head skin pressure stops swimming (stimulus 0.3nA for 30ms excites 10 *tSp*s at time 1.3s; 3) Trunk skin touch initiates swimming (stimulus of 0.3nA, for 5ms excites 2 *RB*s at time 2s); 4) Trunk skin touch during swimming leads to acceleration (stimulus of 0.3nA, for 5ms excites 2 *RB*s at time 4s); 5) Swimming slows and stops spontaneously. **b-d**. Zoomed in view of spike times for neurons on the left side responding to each type of skin stimulation. Inserts show small diagrams of each sensory pathway (see Fig 1c).

addition, swimming can be accelerated by a trunk skin stimulus. Fig 2a (bottom panel) shows the dynamics of period length [19,37,38]. The initial shorter cycle periods can be explained by a higher depolarisation of rostral *dIN*s due to excitation from active *hexN*s lasting for about 1s. This depolarisation leads to post-inhibitory rebound firing after *cIN* spiking. Swimming can also stop spontaneously after post-inhibitory rebound failure.

Our previous model of the tadpole network [19] was based on extensive anatomical and electrophysiological data from experimental studies [18,39]. It included CPG neurons and generated swimming type spiking activities triggered by trunk skin stimulation. Biologically realistic pair-wise connectivity was modelled using a "developmental method" [40] and was then mapped onto a functional model. Swimming initiation was simplified to start after a short delay and stopping was not included. The *CNS model* is enlarged to include decision making for swimming initiation (on stimulated or opposite side) with a long and variable time delay. Thus, the new model generates the appropriate spiking dynamics and reproduces corresponding movements in response to any sequence of sensory inputs.

### *CNS model* design

Here we extend our previous swimming model [19] by adding two populations of sensory neurons (*tSt* and *tSp)*, two sensory pathway neuronal populations (*tIN* and inhibitory *MHR*) and sensory memory neurons (*hexN*s). These populations are responsible for starting and stopping swimming (Fig 1c). The connectivity of these neurons was established using the probabilistic method [41] as well as a novel technique of anatomical and functional similarities between pairs of new and existing populations. In this method we hypothesise that for two similar pairs of populations, the connection probabilities in each pair are inferred from the same probabilistic distribution. The details of two new algorithms for finding connection probabilities are in the Methods.

Our previous swimming model [19] used extensive anatomical data to partially reconstruct the neuronal connectome and physiology of the sensory pathway and CNS neurons (*RB*, *dla*, *dlc*, *dIN*, *cIN*, *aIN*, *mn*). Further work used this connectivity to extract the probability of each pairwise connection in the connectome (probabilistic model [41]).

Crucially, we included new populations of unidentified hindbrain neurons (*hexN*s) which have been proposed to explain the slow build-up of excitation to threshold leading to swimming in response to trunk or head skin touch [24]. The connections between *hexN*s and other populations are specified as follows: (1) To calculate *dla*/*dlc* to *hexN*s connection probabilities we use recordings from irregular spiking midbrain neurons assuming their properties are similar to the *hexN*s (see Methods). (2) We explore the probability space to obtain long lasting irregular *hexN* firing and reliable initiation of swimming. (3) To optimise the probability and strength of *hexN* to *hdIN* connections we use stimulation below the swimming threshold and compare the response of hindbrain *dIN*s with intracellular recordings [24].

**The physiological model** of each single neuron type follows the Hodgkin-Huxley formalism, with parameters that fit the electrophysiological properties of each cell type (details in Methods). Models of trunk touch sensory pathway and CPG neurons are the same as in our previous publications [40–42] except for *dIN*s. The model of *dIN*s is simplified from a multicompartmental model [43] to a single soma/dendrite compartment. For the head touch sensory pathway neurons *tSt* and *tSp* we adjust the *RB* model [42] and parameters of *MHR* and *tIN* models are selected to match the input resistance, threshold current and current-spike frequency curve with experiments. The *hexN* model has two compartments: soma/dendrite and axon [24].

**Pairwise connections of *CNS model*** is based on the connection probabilities between all neurons: $P(i,j)$ is the probability of directed connection from neuron $i$ to $j$, $i = 1,2,\ldots,N$; $j = 1,2,\ldots,N$, and $N$ is the total number of neurons. By sampling from these probabilities we generate a neuron-to-neuron connectome which is represented by the binary adjacency matrix $A(i,j)$, $(i = 1,2,\ldots,N; j = 1,2,\ldots,N)$, where $A(i,j) = 1$ means a directed connection from neuron $i$ to $j$. Fig 3a shows an example connectome with ~ 128K pair-wise connections. Remarkably, when different connectomes were projected to the physiological model of spiking neurons, all of them generated responses corresponding to the ones found in experiments and described below. **Note**: Using different connectomes, we do not change parameters of the physiological model (see Methods).

To estimate the average number of connections in the adjacency matrix from the population with $m_1$ neurons to a population with $m_2$ neurons we consider the Poisson binomial distribution and calculate the mean $\bar{n}$ and the standard deviation $s$:

$$\bar{n} = \sum\nolimits_1^m p_k, \qquad s = \sqrt{\sum\nolimits_1^m p_k(1 - p_k)},$$

where $p_k$, $(k = 1,2,\ldots,m)$ are probabilities of directed connections from one population to another, and $m = m_1{}^*m_2$.

Fig 3b and 3c illustrates the way to derive connections from probabilities. Fig 3b (top) shows the distribution (histogram) of connection probabilities from sensory *tSt* to *tIN* neuronal populations that is skewed towards higher probabilities. Fig 3b (bottom)shows the zoom of the adjacency matrix for these connections. To characterise sparseness of these connections in comparison with the fully connected network we calculate the average probability $\bar{p} = 0.53$ and conclude that about half of possible connections are present.

Similarly, Fig 3c shows the histogram of connection probabilities (top) from mid-hindbrain reticulospinal *MHR*s to *dIN*s that is slightly skewed towards higher values and zoomed connections extracted from the adjacency matrix (bottom). The average probability $\bar{p} = 0.1$ shows that connections (bottom) are rather sparse.

## The role of sensory memory neurons in swim starts

The alternating firing of reticulospinal *hdIN*s on the two sides of the body is what drives swimming [24,44]. In response to weak trunk skin stimuli, a ramp of excitation is recorded in single *hdIN*s [24], (Fig 4a and 4b). This may lead to the start of swimming unpredictably on the stimulated or unstimulated side at variable delays after the stimulus but does not always initiate swimming. Therefore the stimulation *per se* does not define the start or the sidedness of the first motor reactions. Furthermore, *hdIN*s and *mn*s on both sides occasionally produce a few cycles of synchronous left and right firing (synchrony) preceding swimming alternation. Such synchrony has been studied [41,45] but currently appears to have no behavioural function.

What processes underly the decision to swim, the start side, and the delays to the first motor response? How is synchrony avoided? To answer these questions, we study *CNS model* responses to a repeated, weak, left side trunk skin stimulus. We found that the side of first motor response depended on the interplay of three factors: 1) synaptic strengths of excitation from *hexN*s to *hdIN*s; (2) the level of *hdIN* electrical coupling; and (3) the strength of *cIN* inhibition.

We test the hypothesis that the long and variable excitation in *hdIN*s which can lead to the start of swimming derives from the firing of *hexN*s on each side of the hindbrain [24]. In response to trunk touch, *hdIN*s' show a long-lasting wave of post-synaptic excitation that can accumulate to threshold to initiate swimming after long and variable delays. We proposed that the reverberating irregular firing of *hexN*s deliver the excitation to *hdIN*s.

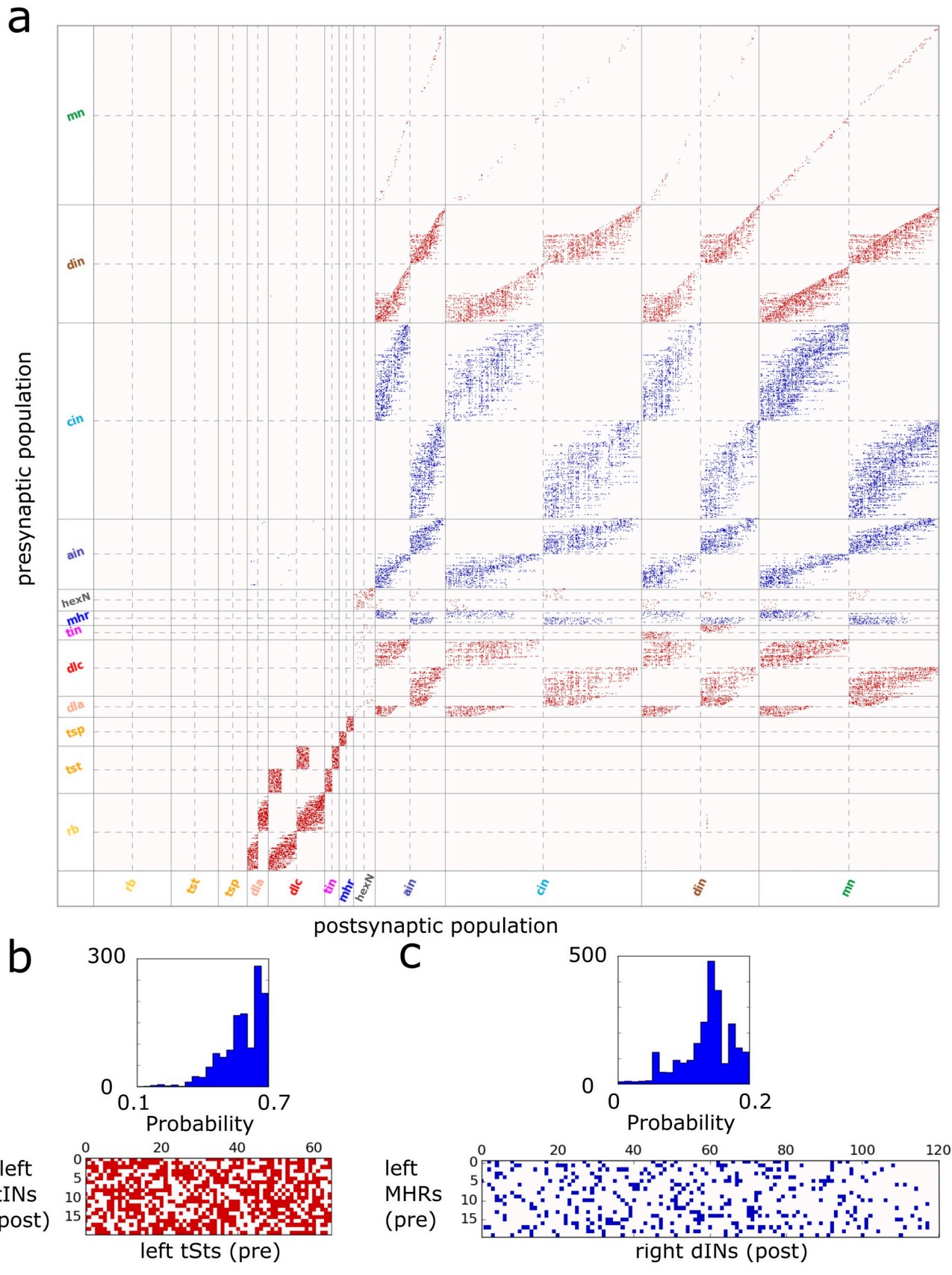

**Fig 3. Connectivity of *CNS model*. a**. Visualisation of the adjacency matrix (connectome), where red are excitatory and blue are inhibitory connections. Rows and columns correspond to pre- and post-synaptic neurons, respectively. There are 12 types of neuronal populations in the *CNS model* and they are separated by solid grey lines. Dashed lines separate the matrix into symmetrical sub-blocks. Within each sub-block vertical and horizontal dotted lines separate the left body side (top rows and left columns) from the right body side (bottom rows and right columns). In each sub-block neurons are ordered according to increasing rostro-caudal position. The matrix describes 128,958 pair-wise connections in one *CNS model* (in 100 models, the mean of total connection number: 128,845.8; s. d. is 1,715.4*)*. **b-c**. Examples of connection probabilities distributions (histograms) and zoomed extraction (lower panels) of excitatory (red) and inhibitory (blue) connections for *tSt->tIN* and *MHR->dIN*, respectively. **(b)** Probabilities $p_k$, ($k = 1,..,m$) for *tSt->tIN* connections are in the range [0.06, 0.66]. Note: this sub-matrix has been extracted from the adjacency matrix $A(i,j)$ and transposed: here columns and rows correspond to pre- and post-synaptic neurons, respectively. The number of connections $m = 691$ and the mean and standard deviation: $\bar{n} = 690.2$, $s = 17.6$ (the mean of non-zero probabilities $\bar{p} = 0.53$ (s.d. is 0.08)—about half of possible connections. **(c)** For *MHR->dIN* connections the probabilities are in the range [0.001, 0.21]. The number of connections $m = 338$, the mean and standard deviation: $\bar{n} = 338.1$, $s = 16.9$, the mean of non-zero probabilities $\bar{p} = 0.1$ (s.d. is 0.07)–sparse connections.

We have inferred the *hexN* activity from the *hdIN* post-synaptic potentials [24] and in addition we have presented examples of candidate *hexN* recordings [9,46] (to properly characterise *hexN* will require many more whole-cell recordings). These showed that some *hexN*s, which are normally silent at rest, fire irregularly following a 1ms trunk skin stimulation and single spike firing of sensory pathway neuron populations at short delays (<13ms; Fig 4c). Based on this indirect evidence, we define a model of the *hexN*s and their connections and propose that a plausible neuronal mechanism of decision making is based on an accumulation of competing excitations to firing threshold in *hdIN*s on each side of the body.

In the *CNS model*, *hexN*s are excited to fire by sensory pathway neurons and extend their brief firing by producing reverberating, irregular firing (Methods) generated via their recurrent ipsilateral and contralateral excitatory connections (Fig 4d and 4g–4j). The *hexN*s excite *hdIN*s and the other CPG neurons, generating a ramp of glutamatergic excitation lasting up to 1.5 seconds on both body sides. This ramp can be seen in the model by averaging *hdIN* potentials in different trials in the case where the swimming is not initiated (Fig 4f) and it qualitatively matches the average of recorded *dIN*s voltages in experiment (below swimming threshold case, Fig 4f). If the excitation from *hexN*s is stronger on one side, the *hdIN*s are recruited to fire by this excitation (Fig 4g) and they recruit the whole electrically coupled *dIN* population on that side [43,47]. This firing recruits other CPG neurons and *mn*s to produce the first motor response. As shown below, a balance in the level of *hdIN* excitation on the two sides is necessary to initiate swimming reliably.

The excitation from *hexN*s to *hdIN*s depends on their irregular firing, their connectivity to *hdIN*s and the strengths of these synaptic connections. All of these factors play a role in the decision making process. To measure the sum of their contributions, we calculate the average potentials $V_L(t)$ and $V_R(t)$ for left and right *hdIN*s (Fig 4g–4j, blue and red lines in the middle panels). If the average potentials $V_L(t)$ and $V_R(t)$ cross the empirically selected threshold level (-27mV) with a time difference more than 4 ms then firing of sufficient right (or left) *hdIN*s occurs to initiate swimming. The dynamics of $V_L(t)$ and $V_R(t)$ and the time difference between threshold crossings are indicators of the swimming starting side and time.

In Fig 4 panels g to j show the results of repetitive simulations with the same stimulus on the left side. For each simulation, we randomly select the connection architecture and strengths. We find that there are four response types depending on the time courses of average membrane potential $V_L(t)$ and $V_R(t)$:

1. **Swimming starts on the unstimulated or stimulated side (Fig 4g):** Both $V_L(t)$ and $V_R(t)$ reach threshold but the time difference between threshold crossings is relatively large. For the example in Fig 4g the potential $V_R(t)$ grows faster, activates right CPG neurons including *cIN*s whose inhibition leads to the significant decrease of $V_L(t)$. However, due to post

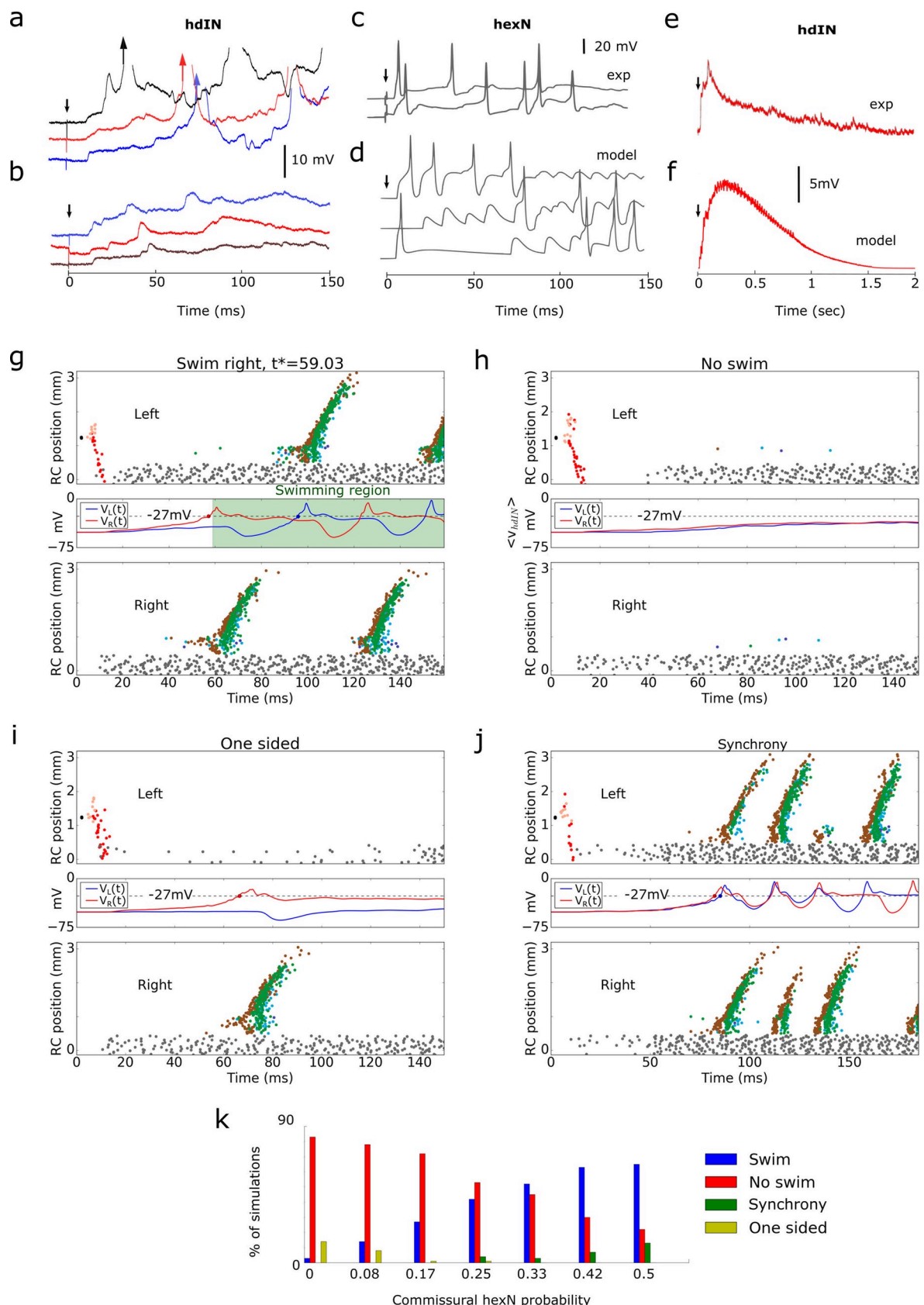

**Fig 4. Activity of *hexN*s and neuronal mechanism of swimming initiation. a-b**. Six recordings of responses to head skin stimulation (black arrow) on the unstimulated side from the same *hdIN* (from Fig 6C and 6D in [24]). **(a)** Excitation ramps to threshold and firing leads to swimming (arrow shows first spike). **(b)** Excitation ramp does not reach threshold. **c.** Two recordings of a possible hindbrain sensory processing *hexN* neuron response to a trunk skin stimulation (at arrow; from Fig 3g in [9]). **d.** Recordings from model *hexN*s to a trunk skin stimulation (at arrow). **e-f.** The averaged ramp of 5 randomly selected model *hdIN*s **(e)** and 5 experimentally (exp) recorded *hdIN*s **(f)** (from Fig 6F in [24]) to a subthreshold trunk skin stimulation (at arrow). In the model, the ramp decays significantly by 1.5 s due to synaptic depression of *hexN*s interconnections. The strength of *hexN*->*hdIN* connection is selected to match the average *hdIN* potential. **g-j.** *CNS model* responses to trunk stimulation: swimming **(g)**, no response **(h)**; one sided activity **(i)**; synchrony **(j)**. Top and bottom subpanels show spike times of active neurons. The central panel shows the averaged voltage dynamics of *hdIN*s on left ($V_L(t)$, blue) and right ($V_R(t)$, red) sides. Coloured dots indicate crossing the threshold (dotted line). In the case of swimming **(g)**, the green area corresponds to swimming activity and the initiation time $t^*$ is the mean of *hdIN* spike times on right. **NOTE**: To make spikes of RB sensory neurons visible, we show them by black colour. **k**. Bar chart shows distribution of four responses for different connection probabilities between *hexN*s on opposite sides. For each probability we run 100 simulations with randomised connectivity and synaptic weights.

inhibitory rebound of left *dIN*s, the potential $V_L(t)$ increases and crosses the threshold after a delay of about 40ms. As a result, swimming starts on the right (unstimulated) side. Similar dynamics of $V_L(t)$ or $V_R(t)$ with fast growing potential $V_L(t)$ can lead to swimming starting on the left (stimulated) side (not shown).

2. **No swimming (Fig 4h):** Neither $V_L(t)$ nor $V_R(t)$ reaches the threshold and swimming is not initiated.

3. **One sided activity (Fig 4i):** Either $V_L(t)$ or $V_R(t)$ reaches the threshold and CPG neurons on one side fire briefly but this activation is not sufficient to initiate swimming. A similar activity pattern, probably equivalent to a simple flexion behaviour, has been seen rarely in experiments.

4. **Synchrony transition to swimming (Fig 4j):** Both $V_L(t)$ and $V_R(t)$ reach the threshold with a small time difference (less than 3ms). In synchrony, CPG neurons on both sides fire simultaneously for 1–15 cycles before switching to swimming [41,45].

One sided and synchronous activity in simulations critically depends on an imbalance in the *hexN* to *hdIN* excitation on the two sides. It was previously shown that the generation of post-inhibitory rebound spikes in *hdIN*s requires a sufficient level of glutamatergic depolarisation [19,41]. One sided activity occurs when the *hexN* to *hdIN* excitation on one side is high enough to generate firing, while on the opposite side excitation is too low for firing on rebound following *cIN* inhibition. Synchrony occurs when the *hexN* to *hdIN* excitation on both sides builds up at similar rates and is strong enough to initiate a simultaneous firing of *hdIN*s on both sides.

The commissural connection probability of left and right *hexN*s ($P_C$) has a strong influence on the motor response (Fig 4k). In experiments, low level trunk skin stimuli lead to swimming in ~ 50% of trials. In simulations of the *CNS models* with $P_C = 0.33$, responses to stimulation are distributed as follows: Swimming 52%; non-swimming 45%; synchrony 3% and one-sided responses 0%. Therefore, we used this optimal value $P_C = 0.33$ in all simulations. Note: for a lower value $P_C = 0.25$, we also observed all types of four responses across random trials.

## Operation of *CNS model*: Swimming initiation

The *CNS model* can reproduce responses to trunk and head skin stimulation [19,24,33]. Stimulating four sensory *RB*s on the left side to fire once (Fig 5a) mimics trunk skin touch (**TT**). This triggers single spikes in the left side population of *dla*s and *dlc*s, which project rostrally to the hindbrain to excite *hexN*s on both sides. The excitation initiates one to two seconds of reverberating firing in *hexN*s and summation of EPSPs in hindbrain *dIN*s reaches firing threshold on the right side. The electrical and chemical coupling between *dIN*s then recruits

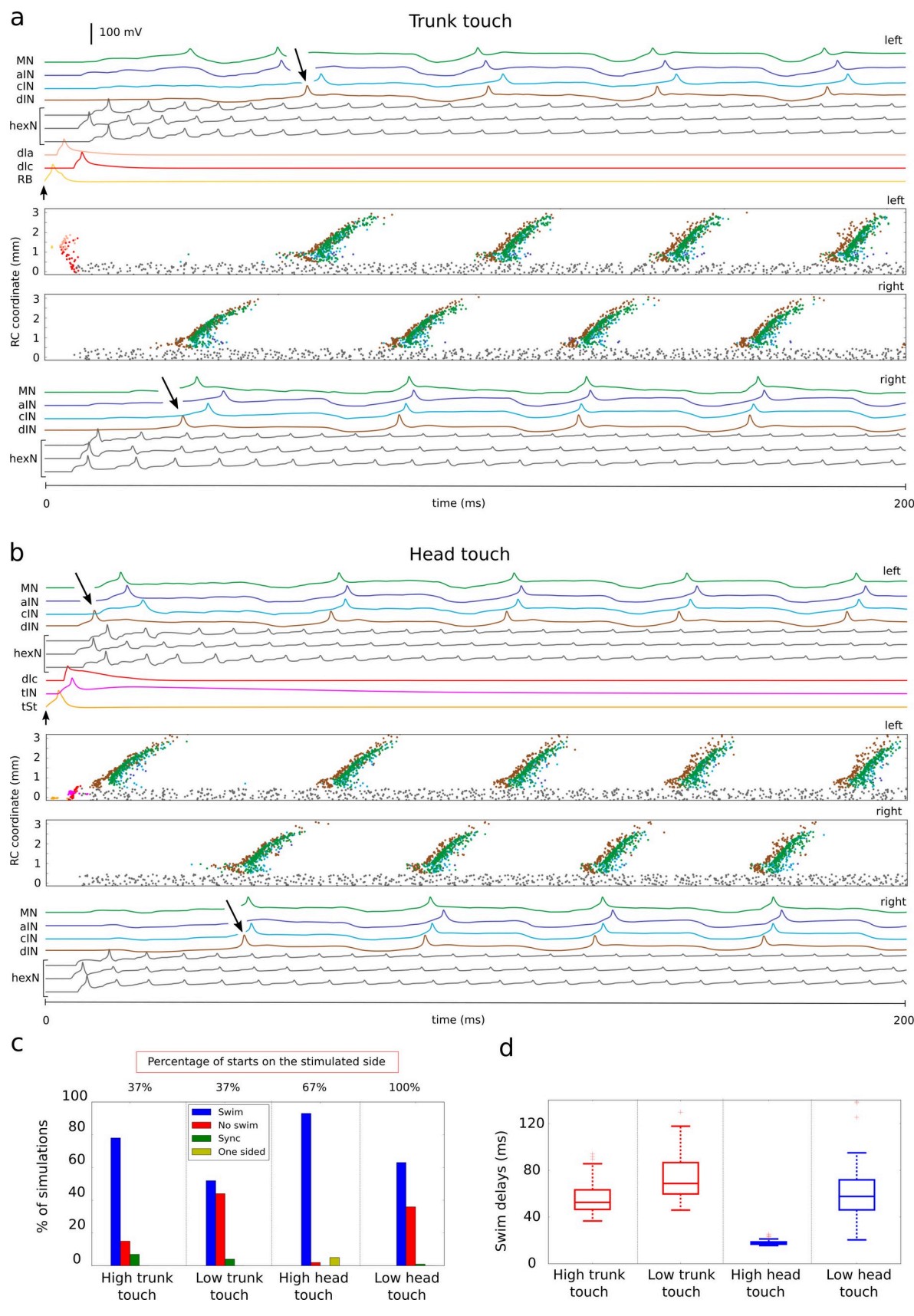

**Fig 5. Swimming initiation in response to trunk and head touch stimulation. a-b**. Neuronal activities on the left and right sides (2 upper and two lower panels, respectively). Central panels show spiking times where the vertical coordinate corresponds to the neuron rostro-caudal position. Other panels show the voltage traces of selected active neurons (one from sensory pathway and CPG populations plus three *hexN*s) and black arrows indicate the first *dIN* spike. **(a)** To mimic trunk touch 4 *RB*s are activated by a brief stimulation at time 0 (arrow, amplitude 0.3nA, duration 5ms). This starts swimming. **(b)** To mimic head touch, 4 sensory *tSt*s fire one spike in response to stimulation at time 0 (arrow, amplitude 0.3nA, duration 5ms). **c-d**. Summary of statistics for simulated responses to high (low) trunk or head skin stimulation. (**c**) Simulation outputs are classified as swim, no swim, synchrony (sync) and one sided. Distribution of outputs is shown for each protocol as well as the percentages of swimming starting on the stimulated side. (**d**) Boxplots of swimming delays for each type of stimulation. Head skin touch (**HT**) is mimicked by stimulation of four sensory *tSt* neurons on the left side to fire once (Fig 5B). This excites *tIN*s and *rdlc*s which fire once to excite *hexN*s on both sides and initiate reverberating firing. Short latency direct inputs from *tIN*s and *hexN*s recruit left dINs which then excite left CPG neurons and initiate swimming.

the whole *dIN* population as well as other CPG neurons and the decision to swim is made. After the initial spikes on the right, CPG neurons on the left receive inhibition from *cIN*s which leads to post-inhibitory rebound (PIR) spikes in left *dIN*s which activate left CPG neurons. Thus, the anti-phase, left-right oscillatory firing of swimming is sustained by PIR and glutamatergic mutual excitation of *dIN*s.

Simulations of the *CNS model* were used to investigate the effects of stimulation strength and our results are in a good correspondence with experimental studies on trunk and head touch initiation with a range of stimulation levels. To mimic **high** (**low**) stimulation [24,35] (also see Methods) we excite four (two) *RB*s for trunk and four (two) *tSt*s for head skin stimulation, respectively. Fig 5c and 5d shows a summary of 100 repeated simulations: **High TT** stimulation leads to swimming with variable delays in 78/100 cases. **Low** level **TT** stimulation generates less reliable swimming with longer and more variable delays in 52/100 cases. For both stimulation levels, the percentage of swimming starts on the stimulated side was 37% (in line with experiments 32% [24]). For **high HT** stimulations the swimming starts very reliably (95/100) with short delays and always on the stimulated side. **Low HT** stimulation leads to less reliable initiation (78/100) with variable delays and 67% of swimming starts are on the stimulated side.

## Operation of *CNS model*: Swimming termination

As the tadpole continues to swim, the frequency slowly drops and swimming can stop spontaneously. In life this is due to adaptive changes in the excitability of active neurons which release purinergic transmitters ATP and adenosine during swimming and reduction of inward currents in *dIN*s [18,37,38,48]. To simplify modelling of this phenomenon, we introduce synaptic depression to recurrent *dIN* NMDA conductances. This leads to the failure of PIR spiking in the dIN population and swimming stops (Fig 6a).

The tadpole usually stops when its head bumps into something and it attaches with mucus from the cement gland [31] (Fig 1a). To model swimming termination by broad head skin pressure we inject current to evoke multiple firing in 10 head pressure sensory *tSp* neurons on one side (Fig 1c). Their firing activates all ipsilateral *MHR*s which project axons to both sides and generate long-lasting GABAergic inhibition of CPG neurons to terminate swimming (Fig 6b). Repeated simulations with different connectivity and synaptic strengths show reliable termination in all cases.

Experiments have shown that the stimulation of single *MHR* after 2.4 seconds of swimming can stop swimming in 78% of trials [31]. To mimic this experiment, we stimulate one randomly selected *MHR* neuron 2.5 seconds after the initiation of swimming in the *CNS model* (Fig 6c). Swimming stopped in 74% of 100 trials with different connectomes. Further simulations showed that stopping reliability increased monotonically with increasing time during a swimming episode due to the decreasing reliability of *dIN* spike generation (Fig 6d, green dot

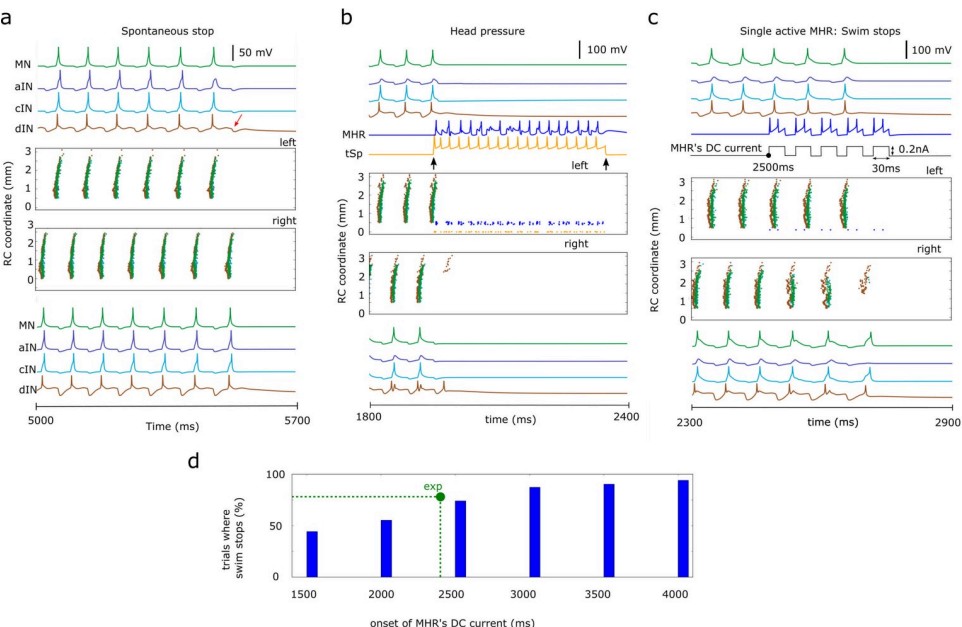

**Fig 6. Swimming termination. a-c**. Neuronal activities on the left (top) and right (bottom) body side for one selected simulation of the *CNS model* near swimming termination. Central subpanels show neuron spike times vs their rostro-caudal position (vertical) and upper and lower subpanels display the voltage traces for different types of CPG neurons. **(a)** Swimming slows before spontaneous termination by synaptic depression at about 5.6 s after stimulation. In the left side *dIN* trace there is no rebound spike after the last reciprocal IPSP (red arrow). **(b)** To mimic the head skin pressure, 10 sensory *tSp* neurons on the left side are injected with a step current (duration 0.4 s, 0.2nA). Both *tSp* and *MHR* neurons fire rhythmically (the mean frequency 32.5Hz), inhibit CPG neurons on both sides and stop swimming. In this case some caudal *dIN*s on the right fire after the start of the inhibition but there is no mn activity. **(c)** A single randomly selected *MHR* is injected with 5 equal DC current pulses starting at 2,500 ms (total duration 0.3 s, each pulse is 30 ms). The single *MHR* fires twice to each pulse, inhibits CPG neurons and this is sufficient to stop swimming. **(d)** Percentage of simulations where swimming was stopped by the activation of a single randomly selected *MHR* vs the onset time of the *MHR*'s current, green dot shows experimentally determined value of stoppings for 2.4 s onset.

shows experimentally determined value and allows to compare experimental and modelling findings).

## Biomechanical *VT model*

Our biomechanical modelling of tadpole swimming is based on the "Sibernetic" software system [49] which has been successfully used to model the 3D body of the *C. elegans* nematode and simulate its swimming and crawling locomotion [30,50]. In Sibernetic, objects and their environment are composed of equally spaced and sized "*particles*" interacting with each other. They can be of different mass to represent liquids or other materials with different densities. Elastic objects are constructed by connecting "*particles*" with "*springs*" with individually defined stiffness and, for muscle tissue, groups of user-defined springs can be instructed to contract in response to incoming motor signals from the *CNS model*. Fluid dynamics calculations are based on the predictive-corrective incompressible smoothed particle hydrodynamics algorithm [51] which considers viscosity, surface tension, density and pressure, as well as gravity and other external forces acting on a liquid.

Reconstruction of the 3D body of the tadpole used photographs, scale drawings and histological sections (Fig 7a–7c). It reproduces its surface shape, length, width, and height and also internal structures: notochord (a longitudinal rod providing stiffness [52]), segmented swimming muscles, undifferentiated belly and cement gland on the head which is adhesive to solid

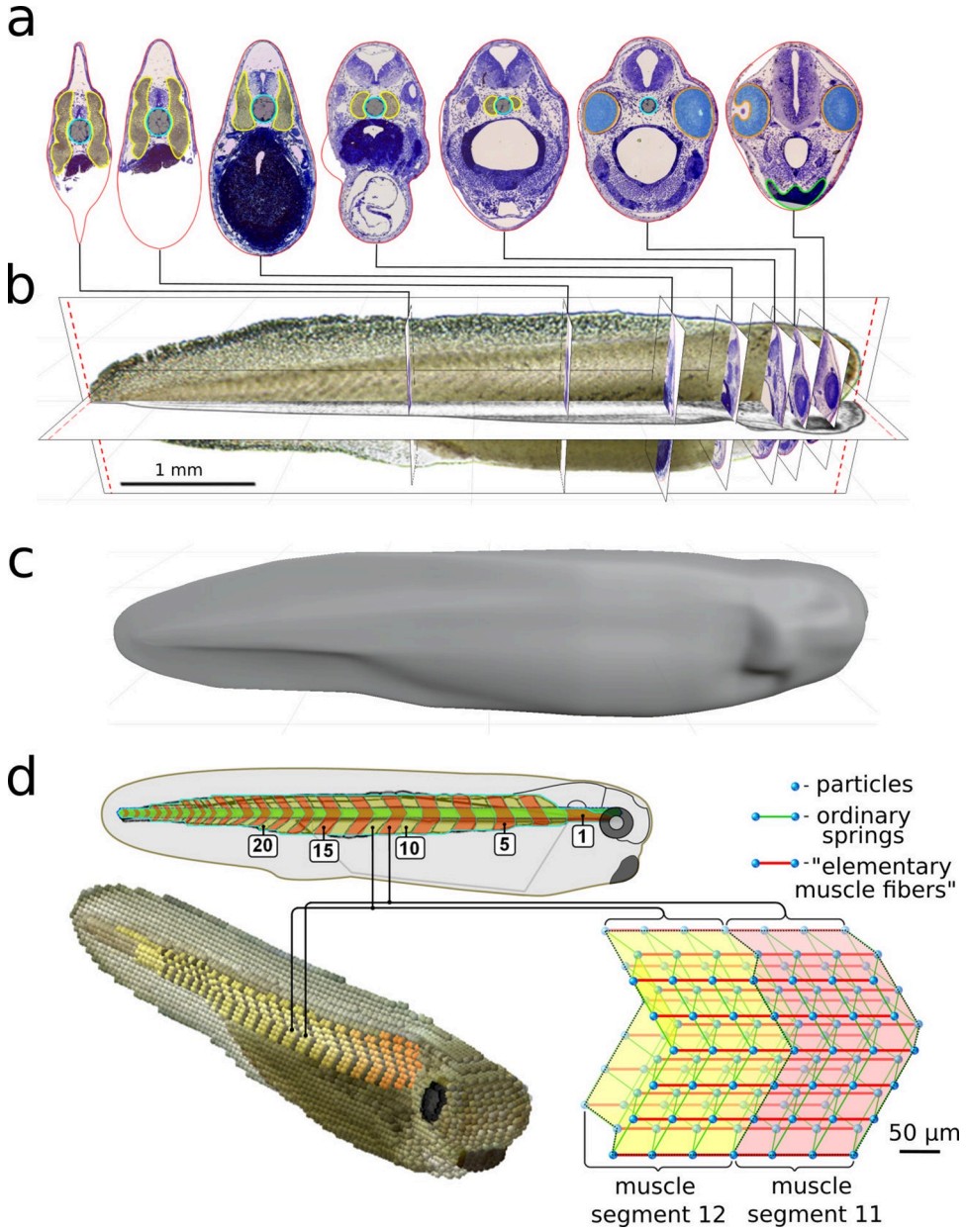

**Fig 7. _VT model_: 3D body reconstruction and swimming in "water". a–c.** Reconstruction of the 3D tadpole body from experimental images and body sections of the real tadpole (using the Blender software). Some body sections at different RC positions from head to tail are shown in **(a)**. **d.** Scale diagram of a real tadpole (top) with the notochord (green) and muscle segments (yellow and orange) with a schematic representation of muscle segments 11 and 12) in Sibernetic (bottom-right). This simplified 3D construction includes particles (blue balls) connected by ordinary springs (green) and "elementary muscle fibers" (red) which can contract in response to motoneuron spiking. **Bottom-left**: Reconstructed 3D tadpole body (_VT model_) in the Sibernetic system. The 3D shape of the tadpole body () was loaded and visualized by Sibernetic to assist in building the _VT model_. The density of the _VT model_ belly is higher (1060 kg/m$^3$) than other tissues (1035 kg/m$^3$). To simulate swimming the _VT model_ is placed into a tank (18.4×3.7×3.7 mm) filled with "water" represented by ≈ 2 million liquid particles with density and dynamical viscosity (measured in the simulation using Stokes' law) equal to those of water at 20˚C. The swimming speed of the _VT model_ ≈ 19.3 mm/s (tadpole length is 5 mm and period of muscle contraction on one body side is 100 ms), whereas the typical preferred speed of a real tadpole is ≈ 21 mm/s. The difference might be due to the skin cilia activity driving surface mucus caudally over the body which is not included to the model [20].

surfaces (Fig 7a–7d). Striving for a balance between model accuracy and computational performance the model body has 14,005 *particles* and is 100 *particles* in length. By adjusting the properties of *"springs"* connecting *"particles"* it is possible to define the stiffness coefficients of various parts of the body. For example, the notochord is more rigid, while the belly region is less rigid and denser [32].

We took care to reproduce the geometry of the segmented muscles that control the body and generate swimming (Fig 7d). Muscles contain longitudinal contractile *"springs"* (elementary muscle fibers: EMFs) with contraction dynamics based on adult frog muscle. *Particles* all over the muscle surface connect with *particles* of adjacent tissues including the notochord. Segments can contract independently when activated by signals from *CNS model* motoneurons (Figs 7d and 8a). To observe swimming the *VT model* of the tadpole body is placed into a tank filled with "water" (Fig 8).

We project a spatio-temporal pattern of motoneuron spiking activity generated by the *CNS model* during swimming to the muscle segments on each side of the VT model body (Fig 8a). Each CNS model motoneuron spike at time $t_{spike}$ causes each EMF within an innervated muscle segment to contract with a force described by a "bell-shape-type" function centred at time ($t_{spike}$+20) ms with summation of forces generated by nearest spikes (Fig 8a and Methods), where 20ms is the delay between motoneuron spike and the maximal EMF force. One side flexes first (contracting muscles are red in video) and the body interacts with the "water" where the velocities of liquid particles are visualized. Alternating flexions then follow and the VT tadpole generates realistic swimming (S1 Video). Key frames from S1 Video show body movements and the corresponding spiking times of motoneurons distributed along the body (Fig 8a). These illustrate swimming initiation and stopping. Initially, the tadpole rests in the middle of the tank near the bottom, the left trunk is stimulated and swimming starts with flexion to the right side (Fig 8b). Swimming stops when the head touches the wall and *VT model* sticks with cement gland mucus (Fig 8c).

In S2 Video the tadpole initially lies on its right side on the tank bottom as it would in life. Skin stimulation on the left leads to muscle contraction on the right side (Fig 8d) and, as swimming continues, the body rotates to a standard dorsal-up, belly-down position. As in real tadpoles the *VT model* moves slowly upwards in the "water" as it swims and, if the tadpole lies on its left or right side and then starts swimming, it is able to reach a dorsal-up position in 2–4 cycles due to a simple ballast effect of the denser belly [32].

## Discussion

The *CNS model* is a neuron-by neuron, biologically realistic reconstruction of the *Xenopus* tadpole hindbrain and spinal cord. It consists of a spiking neuronal network that reproduces the dynamics of neuronal responses seen in whole-cell recordings. The model receives and integrates sensory signals from three skin sensory pathways to make decisions and control the initiation and stopping of swimming.

The spiking activity of motoneurons generated by the *CNS model* is used to activate muscles in the biomechanical *VT model*, which then generates body movements very similar to those of real tadpoles swimming in water and influenced by gravity. Our results present a first attempt to build a detailed biologically realistic 3-dimensional model of a whole animal's body and show how its locomotor behaviour is controlled by CNS neuronal networks.

### Building model circuits

The *CNS model* includes neuronal circuits related to different functions. Finding pair-wise connectivity inside and between circuits is a difficult problem and our results build on a

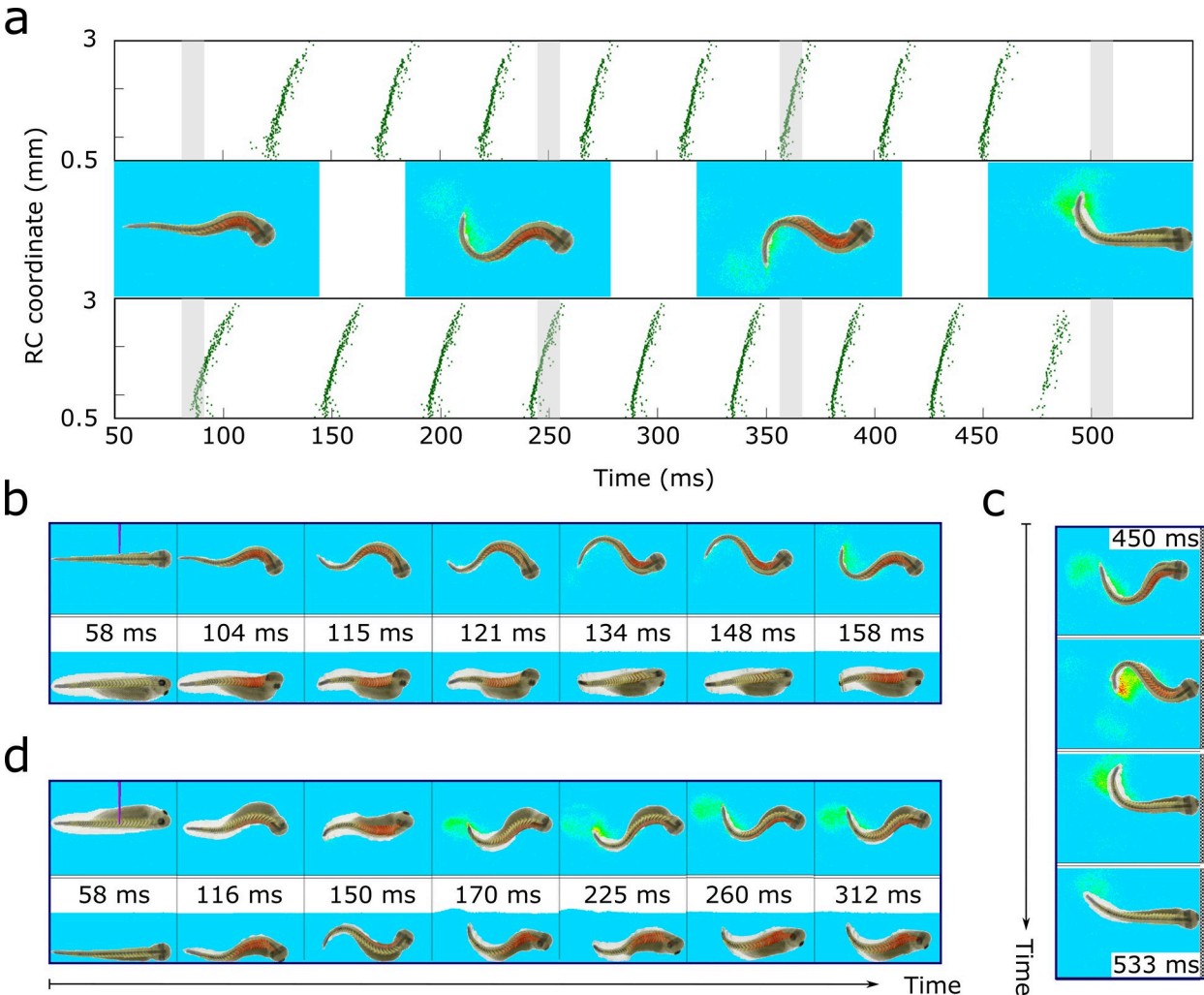

**Fig 8. *VT-model*: Motoneuron spiking, muscle contraction, swimming and stopping. a.** *CNS model* motoneuron spiking times (green dots) and corresponding body positions of *VT model* tadpole during swimming. Horizontal and vertical axes show motoneuron spike time and RC position. Spike related body shapes from video of *VT model* viewed from above are in the middle, and the grey strips show correspondence between spiking and *VT model* swimming (each body shape relates to the middle time point of grey strip). The first frame shows active rostral muscles (red) for the first right flexion. The next two frames show right and left flexions and the last frame corresponds to stopping. Body-water interaction is visualised by the speed of water particle movement (blue-yellow-red corresponds to slow-medium-fast movement). **b–d.** Video frames from the *VT-model* with views from above and one side. **b**. Swimming. Stimulation on the left side at 50 ms is followed by flexion on the right and swimming. Frames are from S1 Video. **c**. Stopping. Sequence of frames from S1 Video show body movements as the tadpole reaches the end of the tank and swimming stopping: the top frame shows the tadpole head approaching the wall; the next frame corresponds to wall contact and cement gland adhesion; last two frames show the attached head and tail moving by inertia. **d**. Effect of gravity. Frames from S2 Video show body rotation under influence of gravity during swimming. In the initial (left) frame the tadpole lies at the bottom on its right body side and the left side is stimulated. The next frame shows the first right side flexion and swimming starts. As it continues, the body rotates and the last two frames show tadpole swimming in a dorsal-up position in the middle of the tank.

growing body of literature in network neuroscience [53–55]. Our "developmental" method [19,40] is based on single cell recording and dye injections for each neuronal population. We consider 2D planes with dorso-ventral and rostro-caudal coordinates on each side of the CNS (the third dimension is negligibly small). Using recorded soma/dendrite locations and axon pathways we allocate dendrites and grow axons in these planes. We generate pairwise synaptic connection with a certain probability at the points of intersection between growing axons and dendrites. Thus, we do not prescribe connections, they appear as a result of a biologically

realistic developmental process. The statistical characteristics of neurons and connections generated by the developmental model are very similar to experimental and anatomical data from real tadpoles.

Our new probabilistic methods combine techniques from probability theory with anatomical data to constrain and define the probabilities of connections for neuronal populations where anatomical evidence is limited. Our anatomical modelling considers the independent Bernoulli variables and similarities in connections between different neuronal populations. This method highlights a path to a general technique for finding pairwise connectivity in complex circuits.

### Shaping connectivity and model parameters

The parameters of our *CNS model* were selected to match extensive physiological data accumulated over decades of research using patch-clamp experiments (see Methods). Most synaptic connections are established using anatomical and/or physiological data by taking into account the "geographical" location of axons and dendrites [56]. For some connections such data were not available, and we used a probabilistic approach. In this approach we define the optimal connection probabilities to match the neuronal activity during swimming. Therefore, swimming activity generated by the *CNS model* was not obtained by parameter adjustments to reproduce the desirable spatio-temporal spiking activity, but appeared as a result of using experimentally defined parameter values for different neuronal types and synaptic connections (see Methods). By incorporating all these experimental details, we have designed a *CNS model* which closely mimics the anatomy, physiology and functionality of real tadpole neuronal circuits. This approach using parameter values from available experimental evidences allows us to increase the model's realism and predictive power.

There are several predictions which follow from our model simulations. For example, connection probabilities and their distributions are predictions for experimental testing. Anatomical measurements of neuronal characteristics and multiple pair-wise recordings can be used to clarify the distribution of connection probabilities. However, currently available measurements and pairwise recordings between sensory pathways and CPG neuronal populations are limited to verify connection probabilities.

### Variable spiking activity but reliable function

The general view on spiking level modelling in neuroscience states that spike trains generated by neuronal circuits are highly variable, while the function of the circuit is stable and reliable [57,58]. The tadpole *CNS model* includes several sources of variability, including variable spike trains in the hindbrain. Despite all these random components this model generates reliable pattern of CPG activity to drive motoneuron firing. The *VT model* was created independently from the *CNS model* using available experimental measurements of physical body characteristics. We adjusted the rigidity coefficient of the notochord, muscles, belly and the rest of the body tissue as well as the maximal contraction force to produce realistic body bending (see Methods). Remarkably, even though the *CNS* and *VT models* were independently designed, mapping the CPG spike trains of the *CNS model* to the *VT model* muscles resulted in reliable swimming behaviour similar to real tadpoles.

### Decision making, variability and unpredictability

Several theoretical ideas on decision making are under discussion [10,59]. However, it is increasingly clear that detailed computational modelling which allows comparison with experimental recordings is rather limited [60,61]. In models of the lamprey nervous system

swimming is reliably initiated by bilateral activation of reticulospinal neurons or by bath gluta-mate applications [62]. However, a complete chain of processing from sensors to reticulospinal circuits to CPG neurons and the start of swimming remains elusive.

Our tadpole *CNS model* with realistic connectivity at neuron- and synapse-level resolution offers an explanation about how the decision to swim is made. We propose that reverberating firing of hindbrain sensory memory *hexN*s produces the long and variable ramps of excitation recorded in hindbrain reticulospinal *dIN*s on both sides which are coordinated by commis-sural *hexN* connections; the ramps of excitation determine the side of first flexion and the initi-ation of swimming. Reciprocal inhibition avoids synchronous firing on both sides. Our modelling results proposes an explanation for the experimental evidence on the long and vari-able delays between stimulation and the start of swimming, the unpredictability of the first flexion side, and the avoidance of antagonist muscle co-activation [24]. Moreover, we propose a plausible hindbrain neuronal mechanism of decision making and swimming initiation: the activities of sensory memory populations on the left and right sides of the body compete to ini-tiate the first body flexion of swimming. We hope that these modelling predictions will be tested by neurobiological experiments.

It is known that there are many similarities in the organization of neuronal circuits controlling locomotion in vertebrates [1] and our simulations shed light on the very basic organisation of the neuronal processes of decision making for movement initiation in all vertebrates. The hatchling *Xenopus* tadpole was chosen for our study because, compared to most animals, its nervous system and behaviour is remarkably simple at this very early stage of its life. In most animals, locomotion is not a simple feedforward process coordi-nated by the CNS but relies critically on motion-dependent proprioceptive sensory feed-back signals (e.g., [63,64], for example, for steering [11,65]). There is at present no evidence for such sensory feedback systems in the newly hatched tadpole and the fre-quency of fictive swimming is very similar to real swimming [66]. The only known sensory receptors which may be activated by tadpole swimming movements are cranial lateral-line receptors responding to water movements [67] and GABAergic cerebrospinal fluid con-tacting neurons in the spinal canal [48]. Related neurons in the lamprey have been shown to respond to mechanical stimulation [68].

## A new approach to biomechanical modelling

The lamprey illustrates the traditional approach to neuro-mechanical modelling where the body is represented by a chain of several hundred segments with flexible joints between them [15]. The motor output of the neuronal model (activity rate of segmental neurons) is trans-formed into two torques which control two-dimensional segmental movement. The interac-tion with water is described by forces acting on each body segment. As a result, the body reproduces two-dimensional movements and swimming emerges as an interplay between internal muscle and external fluid forces. The two-dimensional lamprey model [17] investi-gates how the frequency of tail beating, muscle wave propagation and body curvature sculpt swimming efficiency. The realism of the lamprey model has been increased using chains of coupled CPG oscillators which activate muscles and flex the body [69].

To build our *VT model* we use a different approach. First, we constructed a detailed 3D tad-pole body where the body shape matches multiple photographs (taken from different angles) and cross-sections of the real tadpole. We then built up the tadpole body and surrounding water from many small "particles" (body particles are of different masses). Particles are con-nected to each other by "springs". Spring stiffness is adjusted to reflect the elastic properties of different body tissues. Swimming simulation is based on a computational algorithm that

calculates particle dynamics in 3D space and takes into account viscosity, surface tension, density and pressure, as well as gravity and other external forces [51].

The *VT model* is based on 3D simulation of tadpole movements. It allows us to consider swimming up and down, rotation of the body to the left or right side and other swimming kinematics in a 3D space influenced by gravity. This approach uses advanced and up-to-date modelling techniques and requires simulation of two million of particles and significant computational power with many parallel GPU calculations to provide highly realistic movement. Our particle-based approach, implemented in Sibernetic, uses Eulerian specification of the flow field and is the first application of this technique to swimming in a vertebrate. A similar approach to 3D modelling with Lagrangian specification of the flow field which employs a mesh instead of particles, has been used to study turning during bionic fish swimming [70].

### A new experimental approach to testing neuronal networks

The study of motor control and behaviour is usually centred on neuronal networks for signal processing but it is known that motor behaviour depends on interactions between the nervous and musculoskeletal systems and the external environment [14,15,17]. Our "whole animal" combination of detailed neuronal and biomechanical models proposes to explain the neuronal mechanisms and reproduces realistic swimming in water.

The biological realism of our detailed neuronal *CNS model* and biomechanical *VT model* suggests a protocol for studying how neuronal network activities lead to movement: (1) experimental neurobiological investigation of the nervous system; (2) building a biologically realistic functional CNS model of the connectome; (3) building a virtual biomechanical model to evaluate the behaviour generated.

For example, experimental recordings in an immobilised tadpole will sculpt the *CNS model* and provide questions and hypotheses for simulations. *CNS model* outputs will be used to activate muscles in the *VT model* and generate movements. The output of the *VT model* will provide feedback to experiments and the *CNS model*. Thus, we suggest a new approach to experimental and theoretical studies of movement, where multiple computational experiments with *CNS* and *VT models* can produce new insights on how a nervous system generates appropriate movement in response to input signals from different sensory modalities.

## Methods

### *CNS* model: Anatomical modelling

Each neuron in the *CNS model* is characterized by a type and a RC coordinate (measured in microns from the midbrain-hindbrain border). **NOTE**: In this section we use microns to measure distances but in the RESULTS section we mostly use millimetres.

We have previously shown how to define the locations of tadpole neurons and their connectivity using a "developmental" method [40] that simulated the growth of axons and synaptic formation based on anatomical measurements [19]. We used this method for CPG [*dIN*, *cIN*, *aIN*, *mn*] and sensory pathway populations [*RB*, *dla*, *dlc*].

For the remaining neurons [*tSt*, *tSp*, *tIN*, *MHR*, *hexN*] we use anatomical measurements and uniformly distributed RC coordinates of sensory pathway neurons in the following intervals: *tSt* in $[-50, 0]\mu m$ [33]; *tSp* in $[0, 10]$ $\mu m$ [71]; *tIN* in $[150, 330]$ $\mu m$ [33]; *MHR* in $[400,550]$ $\mu m$ [31]. RC coordinates of hindbrain sensory memory neurons *hexNs* were uniformly distributed in the caudal part of the hindbrain in range $[-130, 500]$ $\mu m$ (hypothetical assumption based on indirect data [24]).

To find connections to and from these cell types we use a probabilistic approach [41], where the directed connection from neuron *i* to neuron *j* was represented by the Bernoulli

variable $X_{ij}$ with probability $p_{ij} = pr\{X_{ij} = 1\}$. We constructed connection probabilities by generating 1000 connectomes using the developmental approach and averaging across connectomes. We used these probabilities to infer the connectivity between populations where there is limited experimental data by hypothesizing anatomical and functional similarities. For example, connection probabilities from trunk skin sensory *RB*s to sensory pathway *dla*s are high, similar to the connection from head skin sensory *tSt*s to sensory pathway *tIN*s. This similarity has been shown by direct pairwise recordings [33]. Using this similarity, we extrapolated from the known connection probabilities of *RB*s to *dla*s in our developmental model [19] and defined connection probabilities for *tSt*s to *tIN*s connections (Algorithm 1).

In many cases some anatomical characteristics like the RC positions of neurons are known. We use this information and define the conditional probability of connection given the distance between neurons (Algorithms 2). For example, for *tIN*s to *dIN*s connections were inferred based on the similarity with *RB*s to *dla*s connections and additionally by considering the RC distances between *tIN*s and *dIN*s.

### Algorithm 1: Anatomical similarity and generalization procedure

We assumed that there is a pair of populations S1 with known probabilities of pairwise connections from the first to second population. We considered another pair of populations S2 with unknown connection probabilities. Also, we assumed that experimental evidence would allow us to propose a similarity between connectivity in pair S1 and in pair S2. After that we use a generalization procedure [40]: we estimate the cumulative distribution of known connection probabilities in S1 based on the developmental model and used this distribution to randomly prescribe a probability to each connection in S2.

### Connections *tSt*s->*tIN*s

The results [33] provided direct evidence of high probabilities of connections from *tSt*s to *tIN*s. We expected that these connections and connections from *RB*s to *dla*s would be similar because in both cases we considered the ipsilateral projections from sensory neurons (*tSts* and *RB*s) to sensory interneurons (*dla*s and *tIN*s) and probabilities of connection from *RB*s to *dla*s are relatively high. Thus, for known probabilities of connections from *RB*s to *dla*s we calculated the piecewise approximation of the cumulative distribution of probabilities and used it to randomly prescribe connection probabilities from *tSt* to *tIN* neurons.

### Connections *tSt*s->*rdlc*s

Similarly, results [72] provided data for similarities of *tSt* to *rdlc* connections and *RB* to *rdlc* connections: both connections were reliable, ipsilateral and from sensory neurons (*tSts* and *RB*s) to *dlc*s. Additionally, in both cases the connections had similar functional properties (provide the sensory signal to initiate swimming). The generalization procedure was then used to find individual connection probabilities.

### Connections *tSp*s->*MHR*s

Previous results [31] provided indirect evidence that these connections have a relatively high probability like connections from *RB*s to *dla*s, and in both cases connections are ipsilateral from sensory cells (*RB*s and *tSp*s). Thus, we assumed similarity and used the generalization procedure to extract the *tSp*s to *MHR*s connection probabilities from *RB* to *dla* probabilities.

## Algorithm 2. Similarity + additional anatomical data

Algorithm 2 uses similarity between connections in two pairs of populations (like in Algorithm 1) and additionally the probability distribution of axonal lengths to find connection probabilities. Assume that S1 is a pair of populations with known probabilities of connections and consider another pair of populations S2 with unknown connection probabilities. Also, we assume that experimental evidence allows us to propose a similarity between connectivity in pair S1 and in pair S2. Additionally, we consider the known probability distribution of presynaptic axons in S1 and assume that the presynaptic axonal lengths in S2 have a similar probabilistic distribution.

The algorithm:

1. Assume that all neurons in each pair (S1/S2) are ordered according their increasing RC coordinate.

2. For each ordered pair of neurons $(i,j)$ in S1 consider the connection probability $P_1$ and the Euclidean distance $L$ between RC coordinates of these two neurons. We then extract the dependence of the probability of their connection on the distance between the two neurons: $P_1 = f(L)$.

3. For each ordered pair of neurons $(m, k)$ in pair S2 find the distance $Q$ between RC coordinates and use the constructed function $P = f(L)$ to find the probability of connection from neuron $m$ to $k$: $P_2 = f(Q)$.

## Connections *tIN*s->*hdIN*s

We know from experiments that *tIN*s have a relatively long descending axons: the mean is 1750 $\mu m$ (s.d. is 480) [33] and similar, for ascending *dla*s axons the mean is 1820 $\mu m$ (s.d. is 470) [72]. Additionally, both *tIN* and *dla* neurons have dorsally located dendrites and fire transiently in response to a touch (either head or trunk skin) and excite CPG neurons [33–35,73]. Thus we expect *tIN*s and *dla*s to have similar connection probabilities with CPG neurons. We use the *dla* to *cIN* connection probabilities to define the function $P = f(L)$, where $L$ is the RC distance between coordinates of *dla*s and *dIN*s and we are taking into account that *dla*s are ascending but *tIN*s are descending neurons; therefore we consider the reflection symmetry for calculation distances. Using this constructed function we find the connection probabilities for *tIN*s -> *hdIN*s.

## Inhibitory connections from *MHR* to CPG neurons

[31] provides indirect evidence that reticulospinal *MHR* neurons connect with high probability to CPG neurons in the rostral part of the spinal cord, sending a powerful inhibitory signal that stops swimming. We use similarities between connections from *dlc*s to all CPG populations and connections from *MHR*s to all CPG populations. There are two reasons for doing this: First, the axons of *MHR* neurons are relatively long, the mean length is 1180 $\mu m$ (s.d. is 350) [72] and the length distribution is similar to the distribution of *dlc* axon length (the mean length is 1190 $\mu m$ (s.d. is 410) [72]). Second, both *MHR* and *dlc* neurons have a similar functional role in the swimming network, they fire transiently in response to sensory input and connect to CPG neurons. We therefore expect that both *dlc*s and *MHR*s to have similar connection probabilities to CPG neuronal populations.

We apply Algorithm 2 to find connections from *MHR*s to each of four CPG populations (*dIN*, *cIN*, *aIN* and *mn*). We construct four functions $P = f(L)$ and use them to establish probabilities of contralateral connection from *MHR* to each of four CPG subpopulations. As *dlc*s are

ascending and *MHR*s are descending neurons, we apply the reflection symmetry to function $P = f(L)$. We know that *MHR* neurons have primary contralateral axons, and about 20% of *MHR*s have secondary ipsilateral axons [31]. Thus, we randomly select 20% *MHR*s in the model and use the same functions $P = f(L)$ to define probabilities of ipsilateral connections from *MHR*s to the corresponding CPG neuronal population.

## Probabilities of connections from *dla* (*dlc*) to *hexN*

Data on *hexN* anatomy is not yet available and electrophyisiological data is limited [9]. However, we recorded the intracellular activity of 33 midbrain interneurons (*mIN*s) in response to trunk skin stimuli near threshold which initiated swimming in 50% of trials. These recordings showed that 45% of *mIN*s (15/33) showed short latency EPSP less than 13ms from stimulation. We assume that: (1) *mIN*s are analogous to the hindbrain *hexN*s because both neuronal populations generate irregular spiking activity in response to stimulation; (2) 45% of *hexN*s demonstrate early EPSPs in the interval [0, 13] ms. It is known that sensory projection neurons *dla* and *dlc* fire only once in response to sensory *RB* cell stimulation and only these neurons spike in 13 ms after the skin stimulation [24,34,74]. Therefore, we conclude that the early *hexN* EPSPs result from synapses made by *dla*s/*dlc*s [9].

We assume that the connection probability from a *dla* (*dlc*) neuron to *hexN* is the independent Bernoulli variable with the probability $p_1$ ($p_2$), respectively. The EPSP in *hexN* can appear as a result of at least one spike incoming from *dla*s or at least one spike incoming from *dlc*s. In addition, from simulations of the spinal cord model [19] we find that the average number of these neurons active after skin trunk stimulation (stimulus to two *RB*s) is $n_1 = 10.4$ *dla*s and $n_2 = 22.9$ *dlc*s.

Also, we assume that the probabilities of direct influence from *dla*s and *dlc*s to *hexN*s are independent and equal:

Pr{*at least one dla spikes*} = Pr {*at least one dlc spikes*}.

Let us consider the event A: {Appearance of EPSP in *hexN* neuron during the time interval [0, 13] ms after the stimulation}.

From the experimental data described above we know: Pr {A} = 0.45.

The probability of $k$ spikes incoming to *hexN* from $n_1$ active *dla*s can be described by the Binominal distribution $B(n_1, p_1)$ (and the Binomial distribution $B(n_2, p_2)$, respectively, for spikes from *dlc*s).

Then, Pr{A} = 1−Pr {*none EPSPs*} =
1−Pr{*none dlas spikes AND none dlcs spikes*} =
1−Pr{*none dlas spikes*}*Pr{*none dlcs spikes*}.
$\Pr\{A\} = 1 - (1 - p_1)^{n_1} \cdot (1 - p_2)^{n_2}$ and $(1 - p_1)^{n_1} = (1 - p_2)^{n_2}$.
From these two equations:
$$p_1 = 1 - \exp\left(\frac{\ln 0.55}{2n_1}\right) = 0.03,$$
$$p_2 = 1 - \exp\left(\frac{\ln 0.55}{2n_2}\right) = 0.013.$$

Thus, we select connection probability 0.03 for *dla*s->*hexN*s connections and 0.013 for *dlc*s->*hexN*s connections.

## Connections *tIN*s->*hexN*s

We hypothesize that *tIN*s make connections to an undefined population of hindbrain neurons *hexN*s [24]. Sensory pathway *tIN*s lie in the rostral hindbrain and have descending ipsilateral axons extending through hindbrain neuron into the spinal cord [33]. Both *dla*s and *tIN*s are excitatory sensory pathway neurons with ipsilateral axons and stimulation of an individual *dla*

can initiate swimming [34]. Thus, we assume similarity between the *tIN*s->*hexN*s and *dlas*->*hexN*s connectivity and prescribe connection probability 0.03 for *tIN*s->*hexN*s connections.

### Establishing the *hexN* recurrent connections

We proposed that *hexN*s are able to generate sustained reverberating firing in response to sensory input as a result of excitatory recurrent connections within the *hexN* population [24]. We assume that *hexN*s are located in a relatively compact longitudinal region of the hindbrain with local connections. The population of motor neurons (*mn*) has mainly short axons and mostly local connectivity within their populations [41]. Therefore, from the connection probability matrix of internal *mn*s connections we select 30x30 sub-matrix (30 first rows and columns because the most-rostral *mn*s are close to the hindbrain region where *hexN*s are presumably located) and use this sub-matrix to prescribe the connection probabilities for inter-neuronal connections of the *hexN* population of 30 neurons. To allow sustained activity within *hexN* population in response to input from *dlas*/*dlc*s it was necessary to multiply these probabilities by a factor of 2.5.

Preliminary simulations of the *CNS model* showed unbalanced excitation of CPG neurons on either side, which would not allow a coordinated initiation of swimming. In addition to ipsilateral connections we added contralateral connections between the left and right *hexN* populations. In addition to the probabilities defined by the 30x30 probability matrix described above, for each pair of *hexN*s on the opposite sides we defined the commissural connection with the probability $p$. We found that the value of this probability contributed to sustaining firing (with similar frequencies) on both sides and to a coordinated initiation of swimming. We therefore explored the space of commissural connection probabilities (see Results) and selected an optimal value $p = 0.33$.

### Connections from *hexN*s to hindbrain CPG neurons

We assumed that the probability of connection from *hexN* to any CPG neuron located in the hindbrain is $p = 0.05$. This probability was adjusted to match the experimentally recorded wave of excitation in *hdIN*s when swimming was not initiated (see Results and Fig 4).

### *CNS* model: Conductance based modelling of neuronal activity

The *CNS model* includes conductance-based single-compartment spike generation neuron models of Hodgkin-Huxley type with synaptic and axonal propagation delays and an additional 2-compartment *hexN* hindbrain neuron class [24]. Synaptic connections are established according the generated adjacency matrix and in addition to the chemical synapses, we also include electrical coupling between *dIN* axons.

The structure and all parameter values of the physiological model, which generates the functional behaviour, are fixed. To run the physiological model we generate a connectome (adjacency matrix of connections) using the probabilistic algorithms and "project" the connectome to the physiological model. The probabilistic algorithms include random components, therefore, if we repeat the algorithm with unchangeable parameter values, the resulting connectome will differ from all others. We generated many (about 1000) connectomes and, remarkably, all of them produced responses corresponding to the ones found in experiments and described in this paper.

To define parameter values of electrical coupling we follow experimental data [19,47] and modelling [43] that suggested that these electrical connections are an important functional property of the *dIN* network. Most parameters of the physiological model are based on available experimental data. For example, synaptic strengths, membrane channel conductance and

neuron capacitances of CPG neurons are mostly based on experimental results and then randomised according to a Gaussian distribution. All simulations were performed using NEURON 7.3 [75] with a fixed time-step of 0.01 ms.

To simulate the activity in the generated connectomes we represent each cell as a single compartment conductance based neuron of Hodgkin-Huxley type. All equations and parameter values for CPG neurons are the same as in paper [41]. Here we provide some key model features.

The equation governing the membrane potential $V_i$ for each neuron $i$ (except *hex*Ns) is

$$C\frac{dV_i}{dt} = I_{lk} + I_{Na} + I_{Kf} + I_{Ks} + I_{Ca} + I_{syn} + I_{gj} + I_{ext}, \tag{1}$$

where $C = 10pF$ is the capacitance of each neuron and corresponds to a density of $1.0\mu F/cm^2$ for a total surface area of $10^{-5}cm^2$. $I_{syn}$ and $I_{gj}$ represent the sum of synaptic and gap junction inputs, respectively. The gap junction input is included only in the *dINs*, based on our previous model [19,43]. The *dIN* gap junction coupling is local and bidirectional by connecting pairs of *dINs* with rostro-caudal positions within $100\mu m$ from each other. For *dIN* with index $i$ the gap junction current is calculated using a simple ohmic relationship:

$$I_{gj} = \sum_{j \in G_i} \bar{g}_{gj}(V_j - V_i).$$

Here $G_i$ is the set of indexes of all *dINs* on the same side of the body as neuron $i$ located within $100\mu m$ from its rostro-caudal position. The gap junction conductance $\bar{g}_{gj} = 0.2nS$, which fits experimental data on *dINs* [19].

The term $I_{ext}$ represents an externally injected current. All cell types except *dINs* have no calcium component ($I_{Ca} = 0$). The leak (*lk*), sodium (*Na*), slow and fast potassium (*Ks* and *Kf*) ion currents are given by Eqs (2–4):

$$I_{Na}(t) = hm^3\bar{g}_{Na}(V_i - E_{Na}) \tag{2}$$

$$I_{Kf}(t) = n_f^4\bar{g}_{Kf}(V_i - E_K) \tag{3}$$

$$I_{Ks}(t) = n_s^2\bar{g}_{Ks}(V_i - E_K) \tag{4}$$

Parameters $E_{lk}$, $E_{Na}$ and $E_K$ correspond the reversal potential for the leak, sodium and potassium channels respectively, and the parameters $\bar{g}_{lk}$, $\bar{g}_{Na}$, $\bar{g}_{Kf}$ and $\bar{g}_{Ks}$ correspond to their maximum conductance. The gating variables $h$, $m$, $n_f$, $n_s$ follow Eqs (5–8), where $X = h, m, n_f, n_s$.

$$\tau_X(V)\frac{dX}{dt} = (X_\infty(V) - X) \tag{5}$$

$$X_\infty(V) = \alpha_X(v)(\alpha_X(v) + \beta_X(v))^{-1} \tag{6}$$

$$\tau_X(V) = (\alpha_X(v) + \beta_X(v))^{-1} \tag{7}$$

$$\alpha_X(V), \beta_X(V) = \frac{A + BV}{C + \exp(\frac{D+V}{E})} \tag{8}$$

A fixed set of parameters $\bar{g}_{lk}$, $\bar{g}_{Na}$, $\bar{g}_{Kf}$, $\bar{g}_{Ks}$ (S1 Table) and $A$, $B$, $C$, $D$, $E$ (S2 Table) define the model of each cell type. Some of these parameters were reported in previous works but we

included them here for completeness. Here is a brief description of the parameters and model used for each cell type.

### RB, dla, dlc, cIN, aIN, mn neurons

Basing on our previous developments [19,41] we selected the same model and parameters for all these neuronal types. This model captures the basic properties shared by all these cell types: firing repetitively to depolarizing current.

### *dIN* neurons

The model *dIN* is based on a recent multi-compartment model [43,76]. For computational limitations we simplified this model by considering only one soma/dendrite compartment. The *dINs* contain a calcium-mediated ion current in the voltage Eq (1) modelled according to the Goldman-Hodgkin-Katz formulation:

$$I_{Ca} = h_{Ca}^2 p_{Ca} z F x \frac{S_{in} - S_{out}\exp(-x)}{1 - \exp(-x)}, \; where \; x = \frac{zFV_i}{RT}$$

Parameter $p_{ca}$ is the permeability of the membrane to calcium ions and parameter $z$ is the calcium ionic valence (+2). $S_{in}$ and $S_{out}$ correspond to the calcium concentrations in and outside of the cell, respectively. Parameters $F$, $R$ and $T$ are Faraday's constant, the ideal gas constant and the temperature, respectively. Parameters of the calcium current are: $p_{ca}$ = 14.25 $cm^3$/$ms$, $F$ = 96485 $C/mol$, $R$ = 8.314$J/(K\,mol)$, $T$ = 300 $K$, $[Ca^{2+}]_i$ = $10^{-7}$ $mol/cm^3$, $[Ca^{2+}]_o$ = $10^{-5}$ $mol/cm^3$. Finally, $h_{Ca}$ is the gating variable associated with the calcium current, which is governed by the gating Eqs (5–8).

### RB/tSt/tSp neurons

The *RB* model is based on previous work and was modified to match the basic electrophysiological properties of these cells [42]. Detailed electrophysiological data characterizing both *tSt* and *tSp* neurons are not known. Since *RB*, *tSt* and *tSp* are all sensory cells we used the same *RB* model for all of them.

### *tIN* neurons

To model *tIN* neurons, we match some of the basic electrophysiological properties of these neuronal type from data in paper [33]. We start from the parameter values of the *aIN* model [42] because both *aINs* and *tINs* have similar firing properties: repetitive firing to current injection at any depolarization level, no firing adaptation, no delayed firing and no after-spike depolarization block. We then adopt the following parameter changes:

- Leak conductance and leak reversal potential are set to match the average input resistance $R_{in}$ = 459 $M\Omega$ and resting potential Vrest = −55mV recorded in experiments [33].

- We use the approach discussed in paper [77] to obtain a qualitatively similar current threshold for firing. Specifically, we decrease by 25mV parameter D defining the rate functions in all ionic channels to obtain current threshold of ~50pA similar to *tINs* (from Fig 6d [33]).

- To obtain a similar dependence of firing frequency in response to current injection (I-f curve) and match the range of firing frequencies of *tINs* we modified maximal conductance of the sodium, fast potassium and slow potassium.

- Parameter A of the $\beta_m$ sodium rate function is decreased by $2ms^{-1}$ to give a better match of the voltage amplitude of spikes during high current injections.

### *MHR* neurons

We model *MHR* neurons to match the electrophysiological properties of these cells[31]. We start from *dlc* model parameters [42] because the *dlcs*' resting potential and input resistance are closer to the average *MHR* values than all other neuronal types in the swimming circuit. Moreover, *MHRs* have similar firing properties to *dlcs*: multiple action potential firing in response to positive current injection and firing adaptation [31,78]. We then adopt the following changes:

- The leak conductance and leak reversal potential are set to match the average input resistance $R_{in}$ = 262$M\Omega$ and resting potential Vrest = −68mV of experimentally recorded *MHRs*. These values were measured from sharp microelectrodes, which record more negative resting potential values than more precise patch electrodes. To correct these imprecise measurements, we select a lower value of the resting potential: *Vrest* = −60mV. This change was suggested by multiple comparisons between neuronal recordings made using both types of electrodes (microelectrodes and patch clamp).

- We decrease parameter D defining the rate functions of all ionic channels by 10mV to obtain a similar current threshold for firing to *MHRs*, following the same approach discussed for *tINs*.

- We match the I-f curve and firing frequencies of *MHRs*. To obtain multiple firing at any level of injected current we lower the spike frequency to physiological values and avoid single-spikes to current injection: Parameter A of the $\alpha_m$ sodium rate function was decreased by $3ms^{-1}$ and parameter A of the $\beta_f$ of the fast potassium rate function was decreased by $0.9ms^{-1}$.

### *hexN* neurons

The model *hexN* is based on the one proposed in paper [24]. Since few experimental recordings of these cells are available, parameters values are randomly chosen from a physiological range to obtain repetitive firing patterns similar to the ones observed in responses to trunk and head touch. The *hexN* model has two compartments: a combined dendrite and soma compartment and an axon compartment. The equations governing the dynamics of both compartments are based on the same Hodgkin-Huxley equations (Eq (1)) with parameter values of *mn* cells [19,24,42]. The parameters for the dendrite/soma and axonal compartments are identical, except that the maximum conductance values of all active channels are increased by a factor of five in the axonal compartment. The capacitance of each compartment is 5pF, and the inter-compartment conductance is 10nS. We use a two-compartment model because single compartment neurons with motoneuron properties are not able to produce persistent rhythmic firing when excited by glutamatergic synapses with NMDA receptors. Strong excitatory input leads to depolarization which stops spiking due to the depolarization block. A more realistic model incorporating two compartments should be free from this problem.

### *CNS* model: Synaptic Currents

The model neuron's synaptic inputs $I_{syn}$ in Eq (1) includes different types of excitatory and inhibitory synapses.

$$I_{syn} = I_{AMPA} + I_{NMDA} + I_{INH} + I_{GABA}$$

Two of these synaptic currents mimic excitatory synapses by releasing glutamate and activating AMPA and NMDA receptors to the post-synaptic neurons. Other two currents mimic inhibitory synapses releasing glycine (INH) and gamma- aminobutyric acid (GABA) and activating inhibitory receptors to the post-synaptic neurons. The type of receptor released in each connection was defined by the anatomical model type depends on the pre-synaptic neuron.

Each synaptic input of a post-synaptic neuron $j$ is given by:

$$I_Y = \sum_i \left\{ w_{i,j}^Y f_Y(V_j) \sum_{s \in S_i(t)} \Delta_Y \left( \exp\left(\frac{s + \delta_{i,j} - t}{\tau_c^Y}\right) - \exp\left(\frac{s + \delta_{i,j} - t}{\tau_o^Y}\right) \right) \right\}, \tag{9}$$

where $Y = AMPA, NMDA, INH, GABA$. The pre-synaptic neuronal type determines the synapse type $Y$. Glycine-releasing inhibitory synapses have presynaptic neuron types $cIN$ and $aIN$; GABA-A–releasing inhibitory synapses have the presynaptic neuron type $MHR$; glutamate-releasing excitatory ones have pre-synaptic neuron types of any of the remaining cell types. Parameter $w_{i,j}^Y$ is the maximum conductance ("strength") of synaptic connection of type Y from neuron $i$ to neuron $j$. When the connectome does not include a connection from $i$ to $j$ this parameter is zero ($w_{i,j}^Y = 0$), otherwise the parameter value is selected according to the type of the pre- and post- synaptic neuron types. In the case that such data is not available, we explored the parameter space to find ranges of values that gave the desired behaviour (see explanation below for each pair of connections).

$S_j(t)$ is the set of spike times of neuron $j$ up to time $t$. Each spike generates a post-synaptic current with rising time constant $\tau_o^Y$ and decaying time constant $\tau_c^Y$. The after-spike increment $\Delta_Y$ is a constant selected for each type of synapse except for the depressing and saturating synapses discussed below. These constants were selected so that the peak of exponentials' difference is 1 in Eq (9), meaning that following a spike the conductance rises to a maximum of $w_{i,j}^Y$. The values selected for the time constants, reverse potential and constant increments are presented in S3 Table (these values are based on previous models established from pairwise recordings [19,42,76]).

## Depressing synapses

As discussed below some features of the tadpole swimming behaviour include synaptic depression in some of the AMPA and NMDA synapses by multiplying the synaptic strength of each synapse by the depression rate $\alpha$ at the occurrence of each pre-synaptic spike (exponentially decreasing strengths [79]). The depressing synapses in the *CNS* model are reported in S4 Table.

## Saturating synapses

Simulations of the neural responses to trunk skin stimulation guided our construction of model synapses. To avoid the transitions from the co-activation of left-right antagonistic muscles (synchrony) to rest while favouring the transitions from synchrony to swimming alternations we included synaptic saturation in the *dIN* to *dIN* NMDA synapses (see Results). Indeed transitions from synchrony to rest are rare in experiments [45] and in this paper the synaptic saturation is modelled via inclusion of variables, rather than a constant, to the after-spike increment $\Delta_{\text{NMDA}}$ defined by:

$$\Delta_{\text{NMDA}}(t) = b \cdot \left( 1 - \left( \frac{c(t) - o(t)}{\sigma} \right) \right),$$

where $b = 1.25$ is the standard increment of NMDA synapses (S3 Table), $\sigma$ is the level of

saturation ($\sigma = 0.025$), and variables $c(t) = \exp((s + \delta_{i,j} - t)/\tau_c^{\text{NMDA}})$ and $o(t) = \exp((s + \delta_{i,j} - t)/\tau_o^{\text{NMDA}})$ appearing in Eq (9).

The function $f_X(V)$ in Eq (9) describes the dependence on the post-synaptic voltage. For $Y = AMPA$, $INH$ and $GABA$ synapses this has a simple linear (ohmic) form:

$$f_Y(V) = E_Y - V,$$

where $E_Y$ is the equilibrium (reversal) potential of the synapse type. NMDA synapses have a non-linear voltage dependence term representing magnesium block and modelled by a sigmoidal scaling factor:

$$f_{NMDA}(V) = (E_{NMDA} - V)(1 + 0.05 \cdot \exp(-0.08 \cdot V))^{-1}.$$

Except connections to and from *hexNs* the delay terms $\delta_{i,j}$ in Eq (9) are dependdent on the rostro-caudal positions of the pre- and post-synaptic neurons. More, precisely, the synaptic delay between two neurons, $\delta_{i,j}$ consists of a constant and distance-dependent part:

$$\delta_{i,j} = \delta_C + \delta_D |P_i - P_j|$$

Here, $P_i$ and $P_j$ are the positions of neurons $i$ and $j$ along the rostro-caudal axis, $\delta_C$ is the constant delay and $\delta_D$ is the speed of synaptic transmission. We set $\delta_C = 1ms$ and $\delta_D = 0.0035ms/\mu m$.

We prescribe constant delays in connections to and from *hexNs* because there is no available data on their rostro-caudal positions. All connections have constant delays $\delta_{i,j} = 1ms$, except the delays for commissural *hexNs* connections, where delays are $\delta_{i,j} = 2ms$ due to the longer distances of commissural axons between *hexNs*.

## Connections types and strengths

For each connection of the adjacency matrix described in section M1 we prescribe a synaptic connection with excitatory or inhibitory active receptors depending on the pre- and post- synaptic neural types. The receptor components and their strengths were prescribed based on several experimental and/or modelling sources (see below). S4 Table summarizes all synapses and their synaptic strength values. Depressing synapses and the citations of the papers used to infer these values are also reported in this table. To mimic synaptic strength variability, the Gaussian noise with the standard deviation 5% of the mean synaptic strength was added to the maximum conductance of all individual synapses, except for the synapses from *hexNs* and *RB*s (see below).

## Synapses from *RBs* to *dlas/dlcs*

Following [19] we included an AMPA component and decreased the mean strength from 8nS to 4nS to distinguish between low/high level stimuli and increase the standard deviation by a factor 10 to increase the trial-to-trial variability and reduce the appearances of synchrony.

## Synapses from *dla/dlc* to *hexN*s

Following [24], we hypothesized that these synapses have both AMPA and NMDA components. There is no available data to infer their strengths. Thus, we explored the space of parameters and found a set of parameters that can explain the long and variable summation of excitation on *dIN* leading to swimming. To reach that we aim: (1) to reproduce a long and variable firing of *hexN*s which follows the trunk touch stimuli and (2) to make swimming start

more reliable on the contralateral side. The selected mean (std) strength of both AMPA and NMDA synapses from *dla* neurons is 7nS (5nS), while from *dlc* neurons is 4.2 (5nS).

## Synapses from *hexNs* to *hexNs*

Following the model [24] we hypothesized mixed AMPA and NMDA components with modified mean strengths (from 1.5nS to 6.5nS for AMPA and from 1.8nS to and 1.4nS for NMDA) to balance the difference in the connectivity between the two modes. The strength is also scaled randomly by a chosen value (0.8, 0.6, 0.4, 0.2 or 0) to introduce trial-to-trial variability. In addition we include synaptic depression with rate 0.993 in both AMPA and NMDA components to stop the irregular firing of the *hexN*s on both sides after ~1.5s in response to trunk skin stimuli. As a result the *hdIN* wave of depolarization generated by *hexN* EPSPs matches the wave of *hdIN* excitation in trials where swimming was not initiated (see Results and Fig 4).

## Synapses from *hexNs* to spinal *CPG* interneurons

Following [24] we hypothesized mixed AMPA and NMDA components with modified mean strengths (from 0.25nS to 1.4nS for AMPA and from 0.1nS to and 0.7nS for NMDA) to balance the reduced number of connections in the model proposed here. Also these strengths are scaled randomly by a chosen value from the set (0.8, 0.6, 0.4, 0.2 or 0) to introduce trial-to-trial variability. Both AMPA and NMDA components include depression with rate 0.98, which was selected to match the *hdIN* wave of depolarization for stimuli below swimming threshold (see Results).

## Synapses from *tSts* to spinal *tINs/rdlcs*

We hypothesized that *tSt* to *rdlc* have the same mean strength as *RB* to *dlc* connections of 4nS because they are both reliable connections from sensory to sensory pathway neurons [19,33]. Experimental work indicates that tSt to tIN connections include mixed AMPA and NMDA components. We include them and fit the connection strengths to match 10–90% rise time, time to peak and duration at 50% of presumed single EPSP from whole cell recordings (S5 Table). The analysis is provided separately for only the AMPA component (obtained by NMDA block via NBOX application) and for mixed AMPA and NMDA components [33]. We fit both sets of data using a python derivative-free optimization algorithm [80] that minimizes the sum of the difference between all the measures in S5 Table. The algorithm's convergence was tested using several initial conditions. Simulated EPSPs are obtained by modelling of a single connection from one *tSt* to one *tIN* under injection a brief step current to the *tSt* to activate a single spike. The results of simulations are ina good agreement woith experimental data. The model measures (optimal values of AMPA and NMDA strengths) are shown in S4 Table.

## Synapses from *tINs* to *hdINs*

Following pairwise whole cell recordings we know that these connections include mixed AMPA and NMDA components [33]. We fit the strength of these connections with the analysis of the 10–90% rise time and time to peak in single EPSPs from whole cell recordings. Data from these connections were only analyzed from mixed AMPA and NMDA components and from pairwise recordings of *tINs* and *dINs*. We are therefore confident that they correspond to single EPSPs from *tINs* to *dINs*. We use the same optimization technique as for finding optimal parameter values in case of *tSt* to *tIN* connections. The desirable values of EPSPs for matching by optimisation are given in S6 Table. Model EPSPs are generated by assuming single synaptic connection from one *tIN* to one *dIN* and by injecting a brief step current to the *tSt* to activate a

single spike. Since the *dIN* voltage dynamics depends on the electrical coupling with other *dINs* in these simulations we included 10 electrically coupled *dINs* with coupling strength 0.2nS.

### Synapses from *tINs* to *hexNs*

We hypothesized that these connections are equal to *dla* to *hexN* connections, since they are both connections from sensory pathway cells to *hexNs* and thus have AMPA and NMDA components with the same strengths.

### Synapses from spinal *CPG* interneurons

Synapses from *cIN*s and *aIN*s have a glycinergic INH component, while the excitatory glutamatergic synapses from *dIN*s and *mn*s have an AMPA component. The strengths of both components are based on previous models that fit whole cell recordings [42]. The AMPA strength of *dIN* to *aIN* connections was reduced (0.6nS to 0.1nS) [19] and *aIN*s were limited to fire shortly after stimulation, as in experiments. Synapses from *dIN*s to *dIN*s include an additional NMDA component with synaptic depression with rate 0.995 to enable the spontaneous termination of swimming activity. The mean initial strength of this component is increased from 0.15nS to 0.2nS to obtain stable alternating oscillations in absence of depression.

### Synapses from *tSps* to *MHRs*

Physiological experiments [31] provide indirect evidence that these connections are excitatory. We hypothesized that they release glutamate (as most excitatory connections in the tadpole) and include mixed AMPA and NMDA components. We selected the mean connection strength to be equal to 5nS and 1nS, respectively.

### Synapses from *MHRs* to spinal neurons

Pairwise whole cell recordings revealed that these synapses release GABA-A [31]. We therefore model a GABA component and use the same mean strength (3nS) as proposed in a previous model [76].

### *CNS* model: Statistical analysis of simulations with high and low level trunk touch and head touch

Here we report *CNS* model responses to trunk touch (TT) and head touch (HT) for the proposed low and high level of stimulations across 100 repeated simulations of the model. For each protocol of stimulation we report the median for the number of activated pathway neurons; the percentage of simulations where swimming is initiated (reliability of swimming start); the median of time delays to the first reaction; and the percentage of initiations where swimming starts on the stimulated side (first flexion on stimulated side). In brackets we show the standard deviation.

- **High TT**: Stimulation of 4 *RBs*. Activated sensory pathway neurons: median is 20.5 *dlas* (2.8) and 46 *dlcs* (6.7). Reliability of swimming start: 78%. Delay: median is 52.4 ms (13.4 ms). First flexion on stimulated side: 37%.

- **Low TT:** Stimulation of 2 *RBs*. Activated sensory pathway neurons: median is 11 *dlas* and 23 *dlcs*. Reliability of swimming start: 52%. Delay: median is 68.7 ms (18.7 ms). First flexion on stimulated side: 37%.

- **High HT**: Stimulation of 4 *tSts*. Activated sensory pathway neurons: median is 31 *dlcs* (2.2) and 15 *tINs* (2.2). Reliability of swimming start: 95%. Delay: median is 17.7 ms (1.9 ms). First flexion on stimulated side: 100%.

- **Low HT**: Stimulation of 2 *tSts*. Activated sensory pathway neurons: median is 21 *dlcs* (3.2) and 6 *tINs* (2.3). Reliability of swimming start: 68%. Delay: median is 57.7 ms (24.5 ms). First flexion on stimulated side: 67%.

### *VT* model: Biomechanical body model

The tadpole body model is represented by an elastic material composed of particles interconnected with springs. Initially, before any bending caused by muscles activity, all body particles belong to a regular hexagonal 3D grid. The volume inside the tadpole body surface shown in Fig 7c is filled with these particles, all of the same size, to form the model body. The distance between adjacent particles in the grid is chosen so exactly 100 of them fit into the longest dimension of its body (from head to tail), which means the total number of particles composing the tadpole body model is 14,005. Depending on the body tissue, we use particles which differ in their properties, mainly in the elasticity of the springs connecting each pair of particles of the same type but also in their mass (which defines the densities of different types of tissues). We consider four types of particles, which form the notochord, the muscle tissue, the belly tissue and the rest of the body tissues (shown in Fig 9).

The shape of the tadpole body is mainly maintained by its most rigid structure–the notochord. It lies along the midline of the body, starting at the head and ending near the tail. In the model it has the shape of a rod (with hexagonal cross-section) with regular structure composed of particles and springs (shown on Fig 9 with black particles).

In the tadpole, body muscles are arranged longitudinally in segments along the body, from head to tail. Left and right muscle segments are symmetric, and the notochord is located between them. In the model the inner sides of the muscles are directly connected to the notochord with springs. Two other types of particles are the belly and the "rest body tissue" (tail fins also belong to this type). The ratio between rigidities of the notochord, muscles, body tissue and belly are 100:20:20:1. The absolute value of spring rigidity was chosen so it provides reasonable elasticity to the overall body model and is neither too rigid nor too soft (spineless). The elastic properties and density of body tissue and muscles are equal; the only difference is that muscles are able to contract in response to activating signals. Left and right muscle bands in the tadpole are segmented into chevron-shaped compartments, which are reproduced in the model quite closely to their real geometry (within the limitations of the regular 3D grid; see upper frame of Fig 7d). In this way we mapped some critical anatomical features from tadpole into our model within Sibernetic. The resolution of the model (and thus the number of particles representing the whole tadpole) can be varied, but we hope that the current choice is near-optimal for modern GPU computational hardware.

Each muscle segment is a 3D elastic object composed of particles and springs. All of its springs, called here elementary muscle fibers (EMFs), are oriented in an anterior-posterior direction (red lines in the right frame of Fig 7d). They are able to contract (move a pair of connected particles towards each other) with the force up to $F_{max}$:

$$\mathbf{F}_{ij}^{\text{EMF}} = F_{max} \cdot \frac{\mathbf{r}_{ij}}{\|\mathbf{r}_{ij}\|} \cdot a, \mathbf{F}_{ji}^{\text{EMF}} = F_{max} \cdot \frac{\mathbf{r}_{ji}}{\|\mathbf{r}_{ji}\|} \cdot a = -\mathbf{F}_{ij}^{\text{EMF}},$$

where $i$ and $j$ are the numbers (indexes) of particles which form the EMF, $r_{ij}$ is a vector between $i$-th and $j$-th particles positions, $r_{ji}$ is an opposite vector (between $j$-th and $i$-th

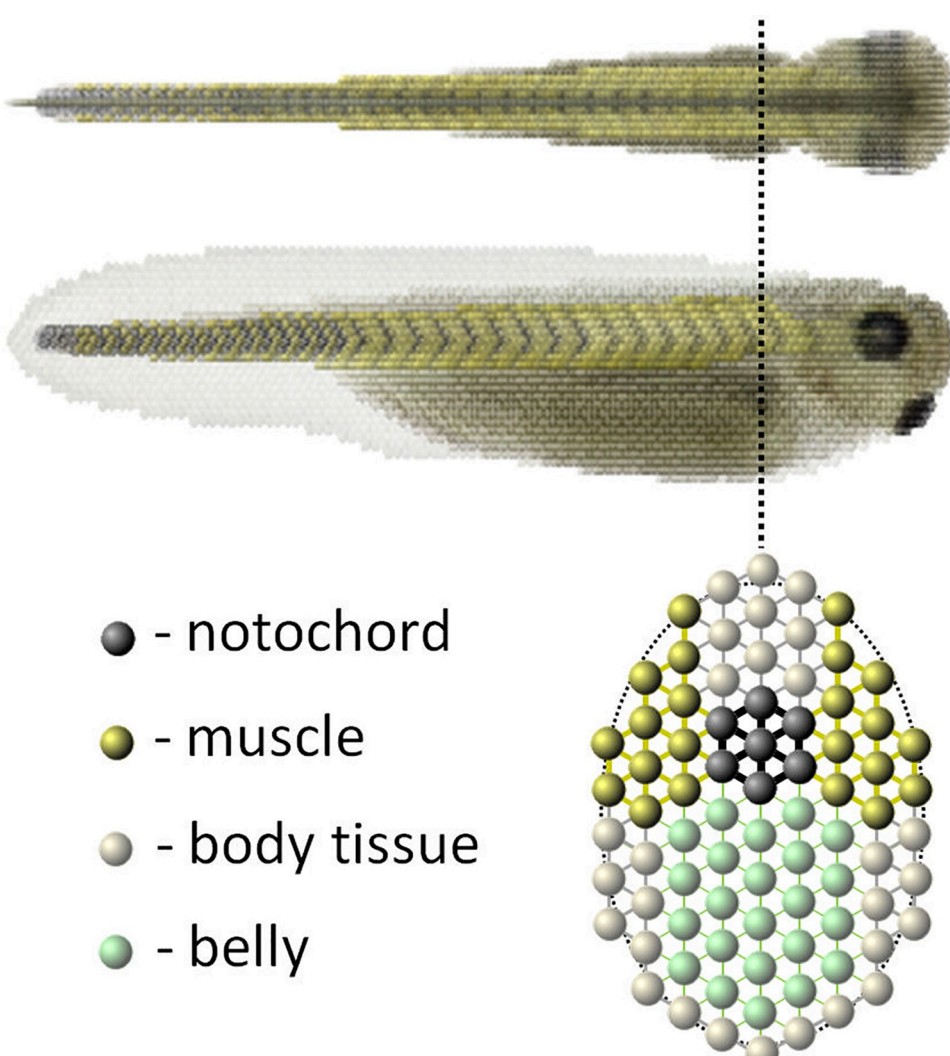

**Fig 9. A simplified view of the particles and springs structure of the VT body cross-section (corresponding to the dotted line).** Four different types of particles are shown in different colours and the springs are shown by connections between particles. Thickness of connecting lines reflects rigidity of springs.

particles positions) and *a* is the activation parameter of the muscle segment being considered (a is between 0 and 1). In the *VT model*, all EMFs within the same muscle segment share the same level of activation. Muscle segments are activated individually and independently of each other. All springs of the tadpole body model, contractile or not, have their resting (relaxed-state) distances, $\mathbf{r}_{ij}^0$, and when the distance between them changes, elastic force arises and to return the spring back following Hooke's law:

$$\mathbf{F}_{ij}^{\text{elastic}} = -k \cdot \frac{\mathbf{r}_{ij}}{\|\mathbf{r}_{ij}\|} \cdot \left( \|\mathbf{r}_{ij}\| - \mathbf{r}_{ij}^0 \right),$$

where *i* and *j* are indexes of particles connected by a spring, *k* and $\mathbf{r}_{ij}^0$ are the coefficient of rigidity and resting distance of the spring, respectively. Since the notochord is the most rigid structure of the body, when muscles on one side of the body contract, the other side (if it is not activated too) becomes stretched, and *vice versa*.

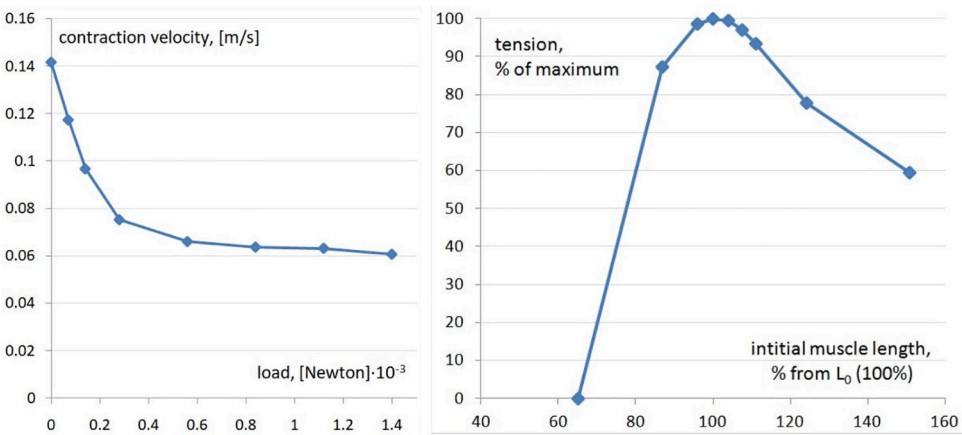

**Fig 10. Basic properties of the muscle model.** Left frame: contraction velocity vs force (load). Right frame: muscle tension vs initial muscle length, L0 is the resting (relaxed) muscle length.

To compare the "muscle properties" of the *VT model* with standard characteristics of real muscles we created a test scene for empirical measuring *VT model* simulations. In these measurements we consider four 4 subsequent muscle segments with the same length (segments from 5 to 8 in Fig 7d, upper frame). We measured two dependencies: (1) The contraction velocity vs force (load) and (2) The muscle tension vs initial muscle length. These dependences are shown in Fig 10 (left and right frames, respectively). We found that in both cases measured dependencies are in a good agreement with typical characteristics of real adult frog muscles [81].

## *VT* model: Transformation of motoneuron spiking times to muscle contraction forces

Each motoneuron's single spike which occurs at time $t_{spike}$ (ms) causes each EMF (elementary muscle fiber) within an innervated muscle segment to contract with the force

$$F(t) = F_{max} \cdot e^{-\frac{(t-20-t_{spike})^2}{50+4 \cdot (t-t_{spike})}}.$$

Where $F_{max}$ was adjusted to provide the most realistic movement.
Summation of forces. We assume:

1. One spike appears at $t_{prev\_spike}$ and it is followed by another spike at $t_{spike}$,

2. The resting activity of the muscle segment at the beginning of the second spike at time $t_{spike}$ is $a_r = e^{-\frac{(t_{spike}-t_{prev\_spike})^2}{300}}$.

Then the force caused by these two spikes is calculated as

$$F(t) = F_{max} \cdot e^{-\frac{(t-20-t_{prev\_spike})^2}{50+4 \cdot (t-t_{prev\_spike})}} + F_{max} \cdot (1 + a_r) \cdot e^{-\frac{(t-20-t_{spike})^2}{50+4 \cdot (t-t_{spike})}}.$$

In the case of more spikes the impact from each new spike is calculated in the same way.

## Supporting information

**S1 Video. Swimming of 3D biomechanical Virtual Tadpole model of a stage 37/38 *Xenopus* tadpole stimulated on the left trunk at 50 ms.** Body movement in the water tank seen from

top and right side views. Muscle activation levels on each side are shown by red colour and numbers at top information panel. The tadpole stops when it hits the tank wall. Speed of water particles shown by colours from light green (slow) to red (fast).
(MOV)

**S2 Video. Body rotation during swimming of 3D biomechanical Virtual Tadpole model of a stage 37/38 Xenopus tadpole.** The tadpole initially lies on its right side on the tank bottom. Skin stimulation on left trunk at 50 ms leads to muscle contraction on the right side and, as swimming continues, the body rotates to a standard dorsal-up, belly-down position. Body movement in the water tank seen from top and right side views. Muscle activation levels on each side are shown by red colour and numbers at top information panel. The tadpole stops when it hits the tank wall. Speed of water particles shown by colours from light green (slow) to red (fast).
(MOV)

**S1 Table. Capacitance, maximal conductance and equilibrium potential of each ionic channel in the model neurons.**
(DOCX)

**S2 Table. Parameters of the rate functions for the Hodgkin–Huxley type neuronal model for all neuron types.**
(DOCX)

**S3 Table. Parameters of the synaptic models for NMDA, AMPA and INH synapses.**
(DOCX)

**S4 Table. Summary of all the synaptic connections between different neuronal populations.**
(DOCX)

**S5 Table. Measures of *tSt* to *tIN* EPSPs in model simulations and experiment.**
(DOCX)

**S6 Table. Measures of single *tIN* to *dIN* EPSPs in model simulations and experiment.**
(DOCX)

## Acknowledgments

We are grateful to Robert Merrison-Hort and Edgar Buhl for valuable feedback on early *CNS* model developments, and to Hannah Ferrario and Joel Tabak for proofreading the manuscript.

## Author Contributions

**Conceptualization:** Andrea Ferrario, Andrey Palyanov, Stella Koutsikou, Steve Soffe, Alan Roberts, Roman Borisyuk.

**Data curation:** Stella Koutsikou, Wenchang Li, Steve Soffe.

**Formal analysis:** Andrea Ferrario, Andrey Palyanov.

**Funding acquisition:** Andrey Palyanov, Stella Koutsikou, Wenchang Li, Steve Soffe, Alan Roberts, Roman Borisyuk.

**Investigation:** Andrea Ferrario, Andrey Palyanov.

**Methodology:** Andrea Ferrario, Andrey Palyanov, Roman Borisyuk.

**Project administration:** Roman Borisyuk.

**Software:** Andrea Ferrario, Andrey Palyanov.

**Supervision:** Alan Roberts, Roman Borisyuk.

**Validation:** Andrea Ferrario, Andrey Palyanov.

**Visualization:** Andrea Ferrario, Andrey Palyanov.

**Writing – original draft:** Andrea Ferrario, Andrey Palyanov, Alan Roberts, Roman Borisyuk.

**Writing – review & editing:** Andrea Ferrario, Andrey Palyanov, Stella Koutsikou, Wenchang Li, Alan Roberts, Roman Borisyuk.

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
