## [Decision Letter · Decision Letter 0]

26 Sep 2021

Dear Dr Ferrario,

Thank you very much for submitting your manuscript "From decision to action: Detailed modelling of frog tadpoles reveals neuronal mechanisms of decision-making and reproduces unpredictable swimming movements in response to sensory signals" for consideration at PLOS Computational Biology. As with all papers reviewed by the journal, your manuscript was reviewed by members of the editorial board and by several independent reviewers. The reviewers appreciated the attention to an important topic. Based on the reviews, we are likely to accept this manuscript for publication, providing that you modify the manuscript according to the review recommendations.

Sincerely,

Matthieu Louis

Associate Editor

PLOS Computational Biology

Samuel Gershman

Deputy Editor

PLOS Computational Biology

[LINK]

Reviewer's Responses to Questions

**Comments to the Authors:**

Reviewer #1: This is a well written computational study describing models of swimming in developing tadpoles. These models are used to test possible mechanisms for decision making in swimming direction. The design and description of the models are rigorous and based upon the literature and previous modelling studies. The models could serve as a tool to explore similar decision-making processes in other species

Minor comments (page numbers refer to the number shown at the bottom of each page and not the PDF page number):

1. Page 2. In the intro paragraph describing the lamprey model, consider another way to describe that model then "The model". It can lead to confusion as to whether you are describing the lamprey model or the model described in this study

2. Page 3. 3rd paragraph. I don't think you can answer the questions with a model but you can certainly propose answers or theories to test

3. Figure 2B-D. There is a mismatch between the brown in the spike times and the brown in the model (which looks more off-gray than brown).

4. Page 8, last paragraph. Replace The estimate with To estimate

5. Page 11. The phrasing Preliminary recordings implies that these are experimental recordings

6. Page 12. Last sentence. By "similar percentage" you mean same percentage of swimming and no swimming response?

7. Can you clarify whether panels g-j from Figure 4 are from the same CNS model or from different CNS models? It would be useful to display the range of left and right membrane potential time courses that preceded each type of event (Swim, No swim, synchrony and one-sided) to get a sense of the sensory memory that usually leads to a specific event

8. "Remarkably, mapping these CPG spike trains to the VT model muscles results in reliable swimming behaviour similar to real tadpole." Was there any parameter fitting in the VT model to make the tadpole model swim like a real tadpole? If there was, then i am not sure it is remarkable. But if the model was created independent of the CNS model output and it worked within the first instance of feeding the VT model with the CNS model, then that is remarkable indeed

Reviewer #2: General comments

The authors should be congratulated on their excellent contribution to the field. They developed a comprehensive neuromechanical model of the tadpole consisting of the central nervous system with 2308 neurons (representing the brain, sensory pathways, CPG, spinal interneurons, and motoneurons) with anatomically realistic interconnections; a biomechanical model of the tadpole body matching body inertial and stiffness characteristics, and a model of interactions of the tadpole with the environment (water). A great advantage of the model is that it’s based on extensive experimental data sets of identified neuronal groups, their interconnections and neural pathways. Another advantage is that author’s bottom-up approach to building the model does not require to do extensive fitting of model parameters to reproduce real motor behaviors. The authors provide a rigorous description of the model of the neural control system and results of its simulated behavior. The major outcome of this study is that the developed model demonstrated probable detailed mechanisms of how motor responses to various tactile stimuli are generated.

I found two issues that authors might want to address to increase impact of their work.

1. Although the authors make a statement throughout the manuscript that the sensory pathways of the model control motor behavior, this statement seems misleading. Yes, sensory pathways originating from skin touch pathways evoke motor responses or stop them, but once a motor response started, it is controlled by feedforward commands from the CPG and motoneurons. In fact, the model lacks motion-dependent sensory signals that have been shown to be critical for performing coordinated movements (e.g., Pearson et al., Trends in neurosciences 29: 625-631, 2006; Bacque-Cazenave et al., J Neurophysiol 113: 1772-1783, 2015). Perhaps the proprioceptive feedback can be incorporated in the model in the future, but this issue should be discussed.

2. Another weakness is that the description of the musculoskeletal system does not have the same rigor and level of details as of the nervous system. Specifically, it seems important to provide details or references of basic properties of the muscle model (the force-length and force-velocity relationships of the contractile component, the force-length relationship of the series and parallel elastic components, history-dependent characteristics, i.e. force enhancement and depression during muscle stretch and shortening), muscle architecture (orientation of muscle fibers with respect to the tendon, muscle fascicle and tendon length, physiological cross-sectional area, muscle moment arms, etc.). All these characteristics affect motor behavior and thus should be described in more details, so other researchers are able to reproduce the simulations.

Specific comments

Figure 1c: The mode of motion control appears to be feedforward without motion-dependent feedback from moving body that modulates CPG activity. This seems counterintuitive given that motion-dependent feedback is critical for maintaining coordinated locomotion (see works of Pearson, Duysens, Prochazka, Akay and others).

Page 6, paragraph 2, “swimming behavior controlled by skin sensory inputs”: controlled or initiated/stopped? The term control would imply that the ongoing rhythmic activity is constantly modulated by skin sensory inputs.

Page 6, paragraph 2, “cycle period lengthens”: What does cause this period lengthening?

Page 8, paragraph 1, “2 sensory, 2 sensory pathway”: ?

Page 9, paragraphs 2 and 3: Is there a way to independently verify the distribution of connections in the animal?

Figure 4k: Is the maximum value on the vertical axis 0.9% or 90%?

Page 18, paragraph 2, “striving for a balance between model accuracy and computational performance…”: How closely the mechanical properties of the model match those of the tadpole's body?

Page 18, paragraph 3, “contraction dynamics based on adult from muscle”: What are contractile properties of the muscles included in the model? Are a series and parallel elastic component are included?

Page 20, paragraph 4, “Using recorded soma and dendrite locations…”: This is an elegant method of finding neuronal connections. But it's not clear if the generated connections are biologically realistic and match the real ones. This should be clarified.

Reviewer #3: Xenopus tadpoles are one of the pioneer model systems for the study of locomotion and as a result a vast amount of detailed experimental data on single-neuron activity and connectivity is available. The neural model of this study is an extension of previous modeling work, incorporates an extensive amount of data, and shows good correspondence with experimental studies. The authors use this model to demonstrate the possible role and feasibility of a sensory memory mechanism to generate experimentally observed variable swim starts. The model is also able to replicate a number of other swimming characteristics.

One novelty of this work lies in the integration of a detailed spiking neural model of locomotion with a biomechanical model of tadpole swimming. The work demonstrates that the spiking neural network is indeed able to generate smooth forward swimming movements and represents a first attempt to build a detailed full-body 3-D neuro-biomechanical model of tadpole swimming. Swimming movements look quite natural by eye and the model seems to sufficiently propel forward in the simulated water, but without a detailed kinematic analysis, it is difficult to assess the correspondence with experimental data.

Nonetheless, I see great value in this work and look forward to future studies building on it.

I have the following comments and concerns.

1) The text is quite difficult to follow and I recommend careful proofreading for clarity, but also for typos and grammar. Generally, I had the impression of carelessness in your text due to the many errors (especially article use, plural/singular, punctuation)

2) I found it difficult to assess the experimental basis for your proposed exINs just from your text. From a quick survey of one of the references you give, I only found some information in reference [37]. I assume your exIN neurons correspond to the proposed hexNs in that paper. Why the name change? From there I gather that these neurons are still unidentified and your model now proposes characteristics of an input population that could provide a long-lasting activity in response to short sensory stimuli. Please revise your text to make clear what experimental data exists and what are your model assumptions.

To illustrate what I mean: You write on page 3 that ”We focus on the critical role of exINs which provide a basic sensory memory of the brief stimulus”. This sounds to me as if these neurons have been identified and characterized in their role. If that is true I couldn’t find it in your references (see also my comment on references). I

In the Methods you state that “few experimental recording of these cells are available”. I might be missing this in ref [37] could you point me and the reader in the right direction?

3) Throughout the text, please double-check your references. I didn’t follow all references, but a few that I checked didn’t seem appropriate. For example:

a. Page 2, 1st para: the selection of one review [1] and one original article [6] seems random to me. The review is focused on spinal locomotor neuronal populations and the article describes a brainstem population that stops locomotion. Could you include more examples?

b. Page 2, 2nd para: “The model of hatchling Xenopus tadpole is built at a unique level of detail compared with existing models”. Reference [7] doesn’t seem at all related to the level of detail that has been used for models of locomotor systems.

c. Page 3, 4rd para: reference {22} doesn’t support the statement that “exINs provide a basic sensory memory”.

d. Page 4, is the thesis publicly available? I couldn’t get access.

4) Page 1, abstract: “reveals that hindbrain sensory memory populations on each side compete to initiate reticulospinal neuron firing and start swimming.” I don’t think you can state it that strongly. Your model predicts that these neurons could play the role, but it doesn’t show anything conclusively.

5) Page 3, 5th para: Caenorhabditis is misspelled, elegans is missing

6) Page 4, 2nd para: you seem to be using hdINs, dINs, and hindbrain dINs interchangeably. I suggest consolidating to a consistent nomenclature. If hdINs and dINs are distinct populations, please clarify that throughout the text and in your schematic.

7) Figure 2, make sure colors between the schematic and recordings match. For example, I found it difficult to find the corresponding recording for dINs since the shades of brown are so different.

8) Figure 2, add (RC) after “rostro-caudal”

9) Page 8, 3rd para: “one of three ways” implies these are alternatives. I think you applied all three instead. If I’m misunderstanding, please explain how you are using these in alternative ways and what the outcomes are.

10) Page 8, 5th para: “Remarkably, when different connectomes were projected to the physiological model of spiking neurons, all of them generated functional behavior that correspond to the ones found in experiments and described below.” Please mention here briefly under what constraints this is true.

11) Page 9, 2nd para: change “formulas” to “equation”, also for other instances of “formula” in the text

12) Figure 3a: label axes with “presynaptic populations” and “postsynaptic populations”

13) Figure 3b: the inversion of pre and postsynaptic axes in comparison with 3a is confusing. Maybe also label with pre and post in addition to your explanation in the text.

14) Figure 4a: are these 6 trials of the same neuron? Six different neurons?

15) Figure 4k: It looks as if none of the exIN probabilities were able to generate all four conditions. Is that correct? Could you comment on that, and I think it's also worth mentioning this in the text.

For Fig 4g-j, what probabilities did you use there?

When you say on page 12 "we selected the optimal value of exIN commissural connection probability p=0.33 to obtain a similar percentage and minimize the percentage of “undesired” one sided and synchrony responses."

What do you mean by that? Select for further investigation? Are all simulations going forward using this probability?

16) Figure 4, legends for c and d, and legends for e and f are swapped

17) Page 22, 3rd para: please tone down language like this: “explains how the decision to swim is made”. Your model offers a possible mechanism and should be seen as a prediction. Experimental work is now necessary to test the hypotheses you put forward.

Same for this sentence just a few lines down “our modelling reveals the hindbrain neuronal mechanisms of decision making and swimming initiation”.

Make sure you edit other sections in the text that overstate your results in this way.

18) Page 23 2nd para: “measured in microns” your axes in Fig 4 for example are labeled with mm. is it micrometer or mm?

19) Page 25, lower third, there seems to be something missing before “ ”

20) The Methods section is missing a description of the biomechanical model

21) Code wasn’t available to me, so I can not review its appropriateness.

**Have the authors made all data and (if applicable) computational code underlying the findings in their manuscript fully available?**

Reviewer #1: **No: **According to the authors, the code will be made available upon publication

Reviewer #2: Yes

Reviewer #3: **No: **Code will be made available at time of publication. Not clear in what form from the "data availability statement"

PLOS authors have the option to publish the peer review history of their article (what does this mean?). If published, this will include your full peer review and any attached files.

Reviewer #1: No

Reviewer #2: No

Reviewer #3: No

Figure Files:

Data Requirements:

Reproducibility:

References:

---

## [Decision Letter · Decision Letter 1]

5 Nov 2021

Dear Dr Ferrario,

Thank you very much for submitting your manuscript "From decision to action: Detailed modelling of frog tadpoles reveals neuronal mechanisms of decision-making and reproduces unpredictable swimming movements in response to sensory signals" for consideration at PLOS Computational Biology. As with all papers reviewed by the journal, your manuscript was reviewed by members of the editorial board and by several independent reviewers. Based on the reviews, we are very likely to accept this manuscript for publication, providing that you modify the manuscript according to the remaining recommendations of reviewer #3.

Sincerely,

Matthieu Louis

Associate Editor

PLOS Computational Biology

Samuel Gershman

Deputy Editor

PLOS Computational Biology

[LINK]

Reviewer's Responses to Questions

**Comments to the Authors:**

Reviewer #2: the authors responded adequately to my comments and suggestions and edited the manuscript accordingly. I have no further comments or concerns.

Reviewer #3: The authors have addressed most of my comments.

The following issues remain.

The text still contains numerous grammatical errors. It’s readable, but I wanted to point this out to make you aware in case you have someone else who could look over the text for you again.

My comment #6 (Figure 1): you added hdIN in the figure legend, but there are no hdIN in the figure. If you use two different terms for them in the text, then they should also be represented in the schematic.

My comment #9 (Page 8, 3rd para) was addressed in the response but the changes were not made in the text.

My comment #10 (Page 8, 5th para): Your response to me was much clearer than what you added in the method section. I suggest you add a similar explanation to the paper.

**Have the authors made all data and (if applicable) computational code underlying the findings in their manuscript fully available?**

Reviewer #2: Yes

Reviewer #3: Yes

PLOS authors have the option to publish the peer review history of their article (what does this mean?). If published, this will include your full peer review and any attached files.

Reviewer #2: No

Reviewer #3: No

Figure Files:

Data Requirements:

Reproducibility:

References:

---

## [Editor Report · Decision Letter 2]

17 Nov 2021

Dear Dr Ferrario,

We are pleased to inform you that your manuscript 'From decision to action: Detailed modelling of frog tadpoles reveals neuronal mechanisms of decision-making and reproduces unpredictable swimming movements in response to sensory signals' has been provisionally accepted for publication in PLOS Computational Biology.

Best regards,

Matthieu Louis

Associate Editor

PLOS Computational Biology

Samuel Gershman

Deputy Editor

PLOS Computational Biology

---

## [Editor Report · Acceptance letter]

1 Dec 2021

PCOMPBIOL-D-21-01493R2 

From decision to action: Detailed modelling of frog tadpoles reveals neuronal mechanisms of decision-making and reproduces unpredictable swimming movements in response to sensory signals

Dear Dr Ferrario,

I am pleased to inform you that your manuscript has been formally accepted for publication in PLOS Computational Biology. Your manuscript is now with our production department and you will be notified of the publication date in due course.

With kind regards,

Zita Barta
